# Climate benefits of proposed carbon dioxide mitigation strategies for international shipping and aviation

Catherine C. Ivanovich[1], Ilissa B. Ocko[1], Pedro Piris-Cabezas[1], and Annie Petsonk[1]

[1]Environmental Defense Fund, 1875 Connecticut Ave NW, Washington, DC 20009

*Correspondence to:* Ilissa B. Ocko (iocko@edf.org)

**Abstract.** While individual countries work to achieve and strengthen their nationally determined contributions (NDCs) to the Paris Agreement, the growing emissions from two economic sectors remain largely outside most countries' NDCs: international shipping and international aviation. Reducing emissions from these sectors is particularly challenging because adoption of any policies and targets requires agreement of a large number of
countries. However, the International Maritime Organization (IMO) and the International Civil Aviation Organization (ICAO) have recently announced strategies to reduce carbon dioxide ($CO_2$) emissions from their respective sectors. Here we provide information on the climate benefits of these proposed measures, along with related potential measures. Given that the global average temperature has already risen 1 °C above preindustrial levels, there exists only 1.0 °C or 0.5°C of additional "allowable warming" left to stabilize below the 2 °C or 1.5 °C thresholds,
respectively. We find that if no actions are taken, $CO_2$ emissions from international shipping and aviation may contribute roughly equally to an additional combined 0.12 °C to global temperature rise by end of century—which is 12% and 24% of the "allowable warming"  we have left to stay below the 2 °C or 1.5 °C thresholds (1.0 °C and 0.5°C) respectively. However, stringent mitigation measures may avoid over 85% of this projected future warming from the $CO_2$ emissions from each sector. Quantifying the climate benefits of proposed mitigation pathways is critical as
international organizations work to develop and meet long-term targets.

## 1 Introduction

There are clear benefits to limiting global average temperature rise to 1.5 °C above preindustrial levels (IPCC, 2018). However, in order to achieve this, carbon dioxide ($CO_2$) emissions likely need to reach net zero around midcentury
(IPCC, 2018). This would require unprecedented changes to energy systems, land use, transportation, infrastructure, and industry worldwide.

Two sectors for which establishing carbon dioxide mitigation policy is particularly complex are international aviation and shipping. The Conference of the Parties to the United Nations Framework Convention on Climate Change (UNFCCC)  in the late 1990s urged that emissions reductions from these sectors be pursued through the UN's
International Civil Aviation Organization (ICAO, established 1944) and International Maritime Organization (IMO, established 1948), respectively (UNFCCC, 1997). While the existence of these UN bodies unites global perspectives for regulation development, this arrangement also requires the agreement of a large number of countries for the adoption of any new policies and targets, a feat much more difficult than if only one or several countries were involved.

While current emissions from international aviation and shipping account for around 4% of global energy-related $CO_2$ emissions (IMO, 2014; ICAO, 2019a; IEA, 2018), emissions from each sector are forecasted to increase anywhere from 200-400% (Lee 2018) and 50-250% (IMO, 2014) by midcentury, respectively, in the absence of effective policy.

Therefore, to support the objectives of the Paris Agreement adopted in 2015, ICAO and IMO have recently announced strategies to reduce carbon dioxide emissions from international aviation and shipping, respectively. As part of a "basket of measures" to address aviation emissions, including a $CO_2$ efficiency standard for aircraft, ICAO adopted a resolution in 2016 to establish a Carbon Offsetting and Reduction Scheme for International Aviation (CORSIA). CORSIA requires States to ensure airlines limit their net emissions of carbon dioxide to 2020 levels, and allows airlines the flexibility to achieve those reductions directly through improved technologies and operations; by reducing emissions outside the sector; and by using fuels that have lower emissions on a life-cycle basis. Efforts are now underway to implement CORSIA, while ensuring that these emissions reductions are not double-counted (once by Paris Parties where the reductions occur, and again by airlines in CORSIA). A long-term goal for international aviation $CO_2$ emissions has been in development since 2008, but ICAO has yet to formally adopt such a target.

On the other hand, IMO announced in 2018 a minimum ambition long-term target of cutting international shipping emissions by at least 50% by 2050 compared to 2008 followed by rapid full decarbonization (IMO, 2018). This long-term target was preceded by various policy options facilitating the reduction of carbon dioxide emissions from the shipping sector. These policies include the Energy Efficiency Design Index (EEDI), which requires increasingly stringent minimum energy efficiency levels for new ships, and the Ship Energy Efficiency Management Plan (SEEMP), which provides an approach for monitoring the energy efficiency of current fleets in use.

While both of these measures—CORSIA and IMO's target—will reduce carbon dioxide emissions from international aviation and shipping, respectively, it is important to analyze the impact that these measures will have on global warming. This information is further important as ICAO's 2019 Assembly considers next steps and IMO revises and reviews its long-term target in 2023 and 2028, respectively.

Several studies have previously quantified the current and future climate impacts of the transport sector, including aviation and shipping. Many studies aggregate the climate impacts of each sector through the use of $CO_2$-equivalence ($CO_2e$) (Lee et al., 2010; Lee, 2018; Eyring et al., 2010; Azar and Johansson, 2012). While the use of this simple metric attempts to describe the global warming intensity associated with the emission of multiple greenhouse gases, it does not account for continuous emissions nor convey warming impacts over time (Ocko et al., 2017). Studies that do investigate climate impacts of these sectors over time often consider the effects of a single emissions pulse or sustained present-day emissions (Fuglestvedt et al., 2009; Berntsen and Fuglestvedt, 2008; Unger et al., 2010).

There are a few studies that have modeled the contribution to warming from future aviation and shipping emissions pathways. One of the earliest estimates was published in the IPCC special report on aviation and its impact on the global atmosphere in 1999; the report estimated aviation's expected business-as-usual (BAU) contribution to warming in year 2050 at 0.05 °C to 0.09 °C. Skeie et al. (2009) also analyzed future warming impacts from shipping and aviation, and included prospective technological improvements in addition to BAU projections. The authors estimated

that aviation's contribution to warming in 2100 will range from 0.11 °C to 0.28 °C, while the shipping sector's contribution will range from -0.01 °C to 0.25 °C, depending on future trends in global economic development. The cooling impact of the shipping sector by midcentury was estimated by Lund et al. in 2012 at -0.02 to -0.04 °C, depending on the assumed emissions scenario. Huszar et al. (2013) estimated that the $CO_2$ emissions from global aviation would produce 0.1 °C warming by the end of the century, and an additional 0.1 °C warming would stem from non-$CO_2$ impacts. More recent estimates from Terrenoire et al. (2019) project that $CO_2$ emissions from the global aviation sector will be responsible for up to 0.1 °C by the end of the century in the absence of mitigation action.

Here we build on these previous analyses by providing information on the climate benefits over time of all proposed and prospective mitigation strategies to date in terms of expected avoided warming compared to BAU projections. We focus our analysis on international emissions as opposed to total emissions (international and domestic). We avoid simple metrics, which do not account for continuous emissions nor convey warming impacts over time, by employing a reduced-complexity climate model. We also account for all climate pollutant emissions from aviation and shipping. We consider the proposed target for international shipping paired with current mitigation policies and more stringent potential revisions, along with various pathways for international aviation which include current CORSIA targets; an extension of CORSIA; and similar targets as those agreed upon by the shipping industry.

Action by both sectors simultaneously is essential because the economics of transport are intertwined. Moreover, the climate impacts of international aviation and shipping are inextricably linked; the success of each industry in its efforts to limit greenhouse gas emissions could drive down the costs of climate solutions and open up new clean fuel supply chains.

## 2 Methods

### 2.1 Business-as-usual emissions from international bunkers

We account for present-day and future emissions of both $CO_2$ and non-$CO_2$ pollutants from international shipping and aviation in our BAU baseline scenarios. Projected emissions over time can be found in Figure 1.

#### 2.1.1 International shipping

International shipping $CO_2$ emissions data for years 2007 to 2030 are taken from the Third IMO Greenhouse Gas Study (IMO, 2014) and for the years 2030 to 2050 are taken from the Update of Maritime Greenhouse Gas Emission Projections (Hoen et al., 2017). Following the 2015 to 2050 growth trend for international shipping, $CO_2$ emissions are linearly extrapolated through year 2100. These projections include estimated $CO_2$ emissions reductions associated with the implementation of EEDI and SEEMP. Because estimations of how these programs will impact the non-$CO_2$ emissions from international shipping are uncertain, they are not included in the baseline emissions profiles for the sector.

For non-$CO_2$ emissions, BAU projections for shipping are taken from the Representative Concentration Pathways Database (RCP Database) using the scenario that most closely represents BAU—RCP8.5 (Riahi et al., 2007). Data are available for the historical and projected shipping emissions of methane ($CH_4$), nitrogen oxides ($NO_x$), carbon monoxide (CO), sulfur dioxide ($SO_2$), black carbon, and organic carbon. These projections do include progressive

reductions to sulfur dioxide emissions associated with the amendments to MARPOL Annex VI, which leads to a 0.5% sulfur dioxide emissions cap in 2020. However, this simplified approach does not consider current Emission Control Areas (ECAs) around the United States, Canada, and the European Union (mandating a cap of 0.1% sulfur) or the potential for ECAs to be declared in additional regions of the world.

**2.1.2 International aviation**

International aviation $CO_2$ emissions data for years 2010 to 2050 are taken from Present and Future Trends in Aircraft Noise and Emissions (ICAO, 2019a). We extrapolate the aviation $CO_2$ emissions for the Low Aircraft Technology and Moderate Operational Improvement Scenario through year 2100 following the 2020 and 2050 trend.

Given that there is a range of reasonable growth patterns for aviation emissions in particular (Lee, 2018; Skeie et al., 2009), and our results depend on this baseline, we ran a set of sensitivity tests to evaluate the influence of different $CO_2$ BAU projection growth patterns on the perceived avoided warming impacts. The two sensitivity tests considered are based on an exponential growth rate pattern through 2100 following the 2005-2050 trend for the high and low demand forecasts as depicted in a previous version of Present and Future Trends in Aircraft Noise and Emissions (ICAO, 2013). These emissions estimates are scaled down to calculate the corresponding Low Aircraft Technology and Moderate Operational Improvement Scenarios proportionally to the latest ICAO forecast (2019a), resulting in declining growth rate patterns in which growth rates follow their 2020-2050 declining trend until plateauing at 0% – as is the case for the low demand scenario. These sensitivity tests are analyzed in addition to the Low Aircraft Technology and Moderate Operational Improvement Scenario noted above, for a total of three analyzed BAU scenarios for aviation. Because of uncertainties associated with how the emissions of non-$CO_2$ pollutants are linked to these different $CO_2$ growth rates, we focused our sensitivity tests on emissions of $CO_2$ in particular. The $CO_2$ emissions profiles used for this sensitivity analysis are shown in Figure 2. We note that all other figures in the paper reflect the Low Aircraft Technology and Moderate Operational Improvement Scenario, which depicts a limited growth pattern for international aviation as this provides a middle of the road estimation.

Aviation emissions data for black carbon and nitrogen oxides are also taken from the RCP Database using scenario RCP8.5. Given that the RCP data includes emissions projections for both international and domestic aviation, we use historical data from the Emissions Database for Global Atmospheric Research (EDGAR; Crippa et al., 2016) to estimate the percent of total emissions from global aviation attributed to international flights (using the most recent data from 2012). Historical international aviation emissions data for sulfur dioxide and carbon monoxide are taken from the EDGAR database, and are linearly extrapolated for each gas in order to match the growth patterns for the other non-$CO_2$ climate pollutant emissions associated with aviation. We estimate international aviation organic carbon emissions based on the RCP black carbon data and using the organic to black carbon ratio (0.49) provided by EDGAR for international aviation emissions (Crippa et al. 2016), again adjusted to reflect only the emissions from international flights. The BAU projections for international aviation sulfur dioxide, carbon monoxide, and organic carbon are added to the business-as-usual scenario including all natural and anthropogenic climate forcings in order to account for their original absence in the RCP8.5 database.

**2.2 Mitigation scenarios**

The mitigation emissions pathways are developed based on a series of agreed upon, proposed, or prospective policy scenarios for international shipping and aviation (Table 1). For international shipping, we analyze two mitigation scenarios: (i) IMO's recently agreed upon minimum ambition mitigation target of reducing carbon intensity by at least 40% below 2008 levels by 2030 and total emissions by 50% below 2008 levels by 2050, followed by full decarbonization of the sector; and (ii) the maximum ambition scenario consistent with pathways to achieve the 1.5 °C target (IPCC, 2018) in which a 40 % reduction in emissions by 2030 is followed by decarbonization of the sector by year 2050. These scenarios assume a linear reduction in emissions between target years, specifically 2015, 2030, 2050, and 2100 for the minimum ambition target; and 2015, 2030, and 2050 for the maximum ambition target.

While these policies are motivated by the intention to reduce emissions of $CO_2$, non-$CO_2$ climate pollutant emissions will likely be impacted as well — although how will depend on the specific methods used to achieve the $CO_2$ targets (Balacombe et al. 2019; Bouman et al. 2017), which are currently undecided. Therefore, we analyze scenarios in which the $CO_2$ mitigation methods do not affect other pollutants, and scenarios in which the $CO_2$ mitigation methods affect other pollutants proportionally; the desire is to capture a range of plausible climate benefits.

Several policy measures have been suggested to reduce carbon dioxide emissions from international aviation. Here we analyze a scenario with emissions reductions necessary in order to maintain a cap on net emissions of international flights to year-2020 levels and four scenarios based on the adoption of CORSIA. The CORSIA-based scenarios include: (i) emissions reductions due to offsets, biofuel use, and improvements in aircraft technology and air traffic management through 2035; (ii) an extension of CORSIA through 2100; (iii) full decarbonization of the international aviation sector by 2100 following CORSIA's completion in 2035; and (iv) full decarbonization of the international aviation sector by 2050 following CORSIA's completion in 2035.

All CORSIA-based scenarios include the maximum potential contribution of improved technology and management (and this maximum potential encompasses the anticipated effects of ICAO's $CO_2$ standard); however, the CORSIA component of the scenario only affects $CO_2$ emissions given that this is an offsetting program. Projections for the $CO_2$ emissions reductions associated with CORSIA through year 2035 are based on the latest list of participating member countries from ICAO (ICAO, 2016b; ICAO, 2019b) and using the Environmental Defense Fund's aviation emissions interactive tool (EDF, 2019). While CORSIA aims to offset international aviation emissions to the point of capping emissions at year-2020 levels, country exemptions to the program allow a small portion of emissions above this cap to remain uncovered. Because no policies currently exist to limit the emissions attributed to these exempt countries, emissions projections for the CORSIA – EXT scenario extend their current growth rate through the end of the century. Projections concerning how both biofuel use and the improvements to aircraft technology and air traffic management will contribute to the future emission of non-$CO_2$ climate pollutants in the aviation sector are very limited and contain high levels of uncertainty, so this tradeoff is not considered in the presented analysis.

### 2.3 Climate model

We employ the reduced-complexity climate model, Model for the Assessment of Greenhouse-gas Induced Climate Change (MAGICC) version 6, because of its widespread and prominent use, and its ability to reliably model climate responses to small forcing changes (Meinshausen et al., 2011a; Ocko et al., 2018). Decades of research have been devoted to improving model parameterizations, and model results demonstrate consistency with sophisticated Coupled Model Intercomparison Project CMIP atmosphere-ocean and $C^4MIP$ carbon cycle models (Meinshausen et al., 2011a).

MAGICC contains a hemispherically averaged upwelling-diffusion ocean coupled to a four-box atmosphere (one over land and one over ocean for each hemisphere) and a carbon cycle model, with an average equilibrium climate sensitivity (ECS) of 3 °C. Between 1765 and 2005, radiative forcings are determined by historical greenhouse gas concentrations (Meinshausen et al., 2011b); prescribed aerosol forcings and land-use historical forcings (National Aeronautics and Space Administration (NASA) GISS model (http://data.giss.nasa.gov/)); solar irradiance (Lean et al., 2010); and historical emissions of ozone precursors (Lamarque et al., 2010). After 2005, radiative forcings are calculated from greenhouse gas emissions (carbon dioxide, methane, nitrous oxide, ozone-depleting substances, and their replacements); tropospheric ozone precursor emissions (carbon monoxide, nitrogen oxides, and non-methane volatile organic carbon); aerosol emissions (sulfate, black and organic carbon, sea salt, and mineral dust); and the indirect effects (first and second) of aerosols.

Whereas radiative impacts of well-mixed greenhouse gases (such as $CO_2$ and methane) are fairly well understood due to our knowledge of gas absorption, aerosol radiative effects are more complex and uncertain. This is due to spatial and temporal heterogeneity complicating observations; a variety of possible microphysical and optical properties based on varying sizes, shapes, structures, mixtures, and humidity levels; and interactions with clouds that can impact the lifetime and brightness of the clouds. Given that aerosols are quite relevant to both the aviation and shipping sectors (e.g. Unger et al., 2010), we include their direct and indirect effects in our simulations, noting that caution must be applied in interpreting the results. Aerosol direct forcings are approximated by simple linear forcing-abundance relationships. The indirect effects of sulfate, black carbon, organic carbon, nitrate, and sea salt aerosols are also included. The effect on cloud droplet size is determined by scaling optical thickness patterns of each species (as described by Hansen et al. (2005)) by their respective emissions. The effect of aerosols on cloud cover and lifetime is modeled as a prescribed change in efficacy of the cloud albedo (for full parameterization details, see Meinshausen et al. (2011a)).

We note that all emissions are treated as surface emissions. Aviation emissions in-flight occur at higher elevations, and this can affect atmospheric chemistry and radiation processes. For example, when sulfate is located above clouds, the radiative efficiency can be halved (less cooling); in contrast, the radiative efficiency of black carbon can be doubled (more warming) when it is located above clouds (Ocko et al. 2012). On the other hand, using more sophisticated climate models that can resolve horizontal and vertical granularities is often complicated by unforced internal variability that makes isolating the climate impact of relatively small radiative perturbations difficult if not impossible (Ocko et al. 2018).

The latest version of MAGICC is not calibrated for inclusion of linear contrails and induced cirrus cloudiness from aviation, phenomena in which water vapor and impurities released in aircraft exhaust form cirrus-like clouds. This is an active area of research and significant progress has been made in recent years to better understand these uncertain processes (e.g. Lee et al. 2009; Schumann et al. 2015; Brasseur et al. 2016; Bock and Burkhardt 2016). In the absence of these parametrizations in MAGICC, we include a sensitivity analysis to show their potential impact on the BAU radiative forcings and temperature responses to aviation.

We use default MAGICC properties with the exception of a few updates to reflect the most recent state of the science. Specifically, we modify methane's radiative efficiency (accounting for shortwave in addition to longwave absorption) and atmospheric lifetime, and tropospheric ozone's radiative efficiency (Etminan et al., 2016; Stevenson et al., 2013). As with climate models of any complexity level, there are limitations in our knowledge of climate and carbon cycle processes, radiative forcings, and especially indirect aerosol effects, which introduce uncertainties within the model. While MAGICC uses several calibration methods to determine its parameters from a large collection of sophisticated models, the comprehensive models will pass along their own uncertainties to MAGICC. Further, due to MAGICC's relative simplicity, parameters are averaged over large spatial scales. This is particularly important to acknowledge as recent literature has demonstrated that radiative forcings associated with the transport sector can differ based on the regional location at which the transport takes place (Berntsen et al. 2006; Fuglestvedt et al. 2014; Kohler et al. 2013; Fromming et al. 2012; Lund et al. 2017; Skowron et al. 2015), particularly for the impact of non-$CO_2$ emissions.

Other major sources of uncertainty stem from the innate inability to accurately project future emissions due to uncertainties in both the human and the climate components of prediction. All mitigation scenarios are compared to an estimated baseline, and the social and economic data utilized in order to inform this estimated baseline cannot be expected to perfectly match the unpredictable nature of human action. Further, the large spatial scales and parameterizations involved in climate modeling contribute to some degree of uncertainty. A full discussion of model uncertainties can be found in Meinshausen et al. (2011a).

**2.4 Climate model simulations**

We run 335 year-to-year integrations from year 1765 to 2100 for a set of 14 different simulations. These simulations are comprised of five BAU pathways and nine mitigation pathways based on current and potential policy scenarios within the international aviation and shipping sectors. For future emissions from sectors other than international aviation and shipping, we use RCP8.5 emissions data, but the climate impacts are subtracted out as described below.

The five BAU scenarios account for the warming impacts due to: all natural and anthropogenic forcings; isolation of the $CO_2$ emissions from international shipping; isolation of the $CO_2$ emissions from international aviation; isolation of the $CO_2$, black carbon, methane, nitrogen oxides, sulfur dioxides, organic carbon, and carbon monoxide emissions from international shipping; and isolation of the $CO_2$, black carbon, nitrogen oxides, sulfur dioxide, organic carbon, and carbon monoxide emissions from international aviation. The nine mitigation simulations account for the future emissions pathways for the nine policy scenarios outlined in Table 1.

In order to isolate sector emissions in each BAU and mitigation scenario, we subtract the total emissions of all gases and aerosols associated with each sector from the total RCP8.5 emissions of all gases and aerosols in the all-forcing scenario driven by all natural and anthropogenic forcings (Eq. 1). The annual average mean surface temperature changes from these emissions profiles are subtracted from the temperature changes in the all-forcings scenario in order to determine the contribution to future temperature change from each sector (Eq. 2). It is important to note that the background temperature response to other forcings (anthropogenic and natural) can affect the temperature responses to shipping and aviation. Therefore, even though they are ultimately subtracted out in our calculation, they do impact our results, and uncertainties in BAU emissions from other sectors and the resulting temperature effects need to be acknowledged.

$$Emissions_{all-forcings\ without\ sector} = Emissions_{all-forcings} - Emissions_{sector} \qquad (1)$$

$$\Delta T_{sector} = \Delta T_{all-forcings} - \Delta T_{all-forcings\ without\ sector\ emissions} \qquad (2)$$

The comparison of each sector's baseline scenario to its respective mitigation scenarios are analyzed independently from other potential mitigation efforts that may occur in the future. Thus, isolating the temperature impacts of a given mitigation scenario does not mandate that all other anthropogenic emissions continue unabated. The same methodology can be used to isolate temperature changes due to individual gases or aerosols for each sector.

## 3 Results

### 3.1 BAU climate responses

Both the shipping and aviation sectors emit a combination of warming and cooling climate pollutants and precursors. The net temperature impact depends on the magnitude of emissions, the radiative efficiencies, and the atmospheric lifetimes of the individual species. $CO_2$ builds up in the atmosphere over time and thus its forcing increases gradually with constant emissions, whereas short-lived species such as all aerosols would yield constant annual forcings with constant emissions. Given that we are analyzing climate impacts of future emissions from international aviation and shipping (year 2020 through 2100), the near-term radiative forcings (defined as the forcing at the tropopause after stratospheric temperature adjustment) are dominated by non-$CO_2$ pollutants and the long-term radiative forcings are dominated by $CO_2$ (Figure 3).

The net radiative forcing for international shipping is -47 mW m$^{-2}$ in 2020 and +48 mW m$^{-2}$ in 2100. The shift from negative to positive is due to the large increase in $CO_2$ emissions and their accumulation over time in the atmosphere. A considerable amount of the positive radiative forcing from $CO_2$ emissions in 2100 (+127 mW m$^{-2}$) is offset by a relatively large negative radiative forcing in 2100 from $NO_x$ emissions (-66 mW m$^{-2}$). Net radiative forcing due to $NO_x$ emissions is a combination of negative and positive radiative forcings from indirect effects; negative forcings arise from reductions in methane, production of nitrate, and nitrate's effect on clouds, and positive forcings arise from production of tropospheric ozone. Indirect aerosol effects from all species yield a radiative forcing of -32 mW m$^{-2}$ in 2100.

Radiative forcings derived in this study from shipping emissions of $CO_2$ and $NO_x$ are consistent with the literature. Previous estimates of $CO_2$'s present-day (early 2000s) impact range from +26 to +43 mW m$^{-2}$, corresponding to emissions of 500 and 800 $TgCO_2$ yr$^{-1}$ (Eyring et al. 2010). This is consistent with this analysis when accounting for the anticipated growth in $CO_2$ emissions of more than fivefold by 2100 since the early 2000s (IMO, 2014). Previous studies estimate radiative forcings from $NO_x$ that range from +8 to +41 mW m$^{-2}$ for indirect effects on tropospheric ozone (compared to our value of +25 mW m$^{-2}$ in 2100) and -56 to -11 mW m$^{-2}$ for indirect effects on methane (compared to our value of -22 mW m$^{-2}$ in 2100) for present-day emissions around 2.9 to 6.5 TgN yr$^{-1}$ (we assume $NO_x$ emissions of 5.6 TgN yr$^{-1}$ in year 2100) (Eyring et al. 2010). For $SO_2$ emissions from shipping, previous studies estimate direct radiative forcings from -47 to -12 mW m$^{-2}$ due to production of sulfate; our estimate is -14 mW m$^{-2}$ in 2100 from emissions that are lower (2.0 TgS yr$^{-1}$) than present-day values in the literature (3.4 to 6.0 TgS yr$^{-1}$) (Eyring et al. 2010). Our estimate of direct radiative forcing from black carbon (+5 mW m$^{-2}$ in 2100 from emissions of 0.2 TgBC yr$^{-1}$) is slightly higher than estimates in the literature (+1.1 to +2.9 mW m$^{-2}$ in 2000/2005 from emissions of 0.05 to 0.3 TgBC yr$^{-1}$) (Eyring et al. 2010). Indirect effects of aerosols have enormous ranges in estimates in the literature (Righi et al. 2011), but we note that our estimate appears to be on the lower end.

The net radiative forcing for international aviation emissions (note: not including impacts on contrails and cirrus clouds) is -1.4 mW m$^{-2}$ in 2020 and +62 mW m$^{-2}$ in 2100. Although radiative forcings are smaller for $CO_2$ for aviation compared to shipping, due to slightly less emissions, there are proportionally less emissions of the negative forcing precursors $NO_x$ and $SO_2$, yielding higher net radiative forcing from aviation. As with the shipping forcings, the large $CO_2$ radiative forcing in 2100 (+87 mW m$^{-2}$) is partially offset by the strong negative forcing from $NO_x$ emissions (-24 mW m$^{-2}$). Indirect aerosol effects from all species yield a radiative forcing of -10 mW m$^{-2}$ in 2100.

Estimates of present-day radiative forcing from aviation in the literature include both domestic and international emissions, whereas our estimates of future radiative forcings exclude domestic travel. Our estimates of radiative forcing from $CO_2$ emissions are in agreement with previous estimates when accounting for different emissions inputs (such as +87 mW m$^{-2}$ in 2100 from emissions of 3670 $TgCO_2$ yr$^{-1}$ compared to +28 mW m$^{-2}$ in 2005 from emissions of 641 $TgCO_2$ yr$^{-1}$ in Lee et al. (2009)). Our estimates for radiative forcings from $NO_x$, $SO_2$ (direct), and black carbon (direct) are slightly smaller than what is presented in the literature, despite larger emissions projected for year 2100 compared to present-day, but there are large uncertainties associated with these estimates and a low level of scientific understanding (Sausen et al., 2005; Fuglestvedt et al., 2008; Lee et al., 2009). For example, Brasseur et al. (2016) estimate +6 to +37 mW m$^{-2}$ for indirect effects of $NO_x$ emissions on tropospheric ozone (compared to our value of +11 mW m$^{-2}$ in 2100) and -8 to -12 mW m$^{-2}$ for indirect effects on methane (compared to our value of -8 mW m$^{-2}$ in 2100). Gettelman and Chen (2013) conduct a more sophisticated assessment of the climate impact of aviation aerosols than what is presented here, and report an estimate of -46 mW m$^{-2}$ from combined sulfate direct and indirect effects; this is considerably larger than our estimate of -3 mW m$^{-2}$ in 2100.

Radiative forcings directly impact temperatures – a net positive forcings has a warming tendency, and a net negative forcing has a cooling tendency. Figures 4 and 5 show the temperature responses over time to projected emissions from both sectors. Given that the ambition for the proposed and agreed upon mitigation policies within the international

shipping and aviation sectors is based on the need to cut $CO_2$ emissions from each sector, we isolate the temperature impacts from the $CO_2$ emissions in addition to the net effect from all emitted climate pollutants.

Figure 4a shows the impact of future international shipping emissions (beginning in 2020) on surface air temperature change throughout the 21st Century. In the year 2020, the impact on temperature represents the contribution from that years' worth of emissions only, and then every year forward represents the cumulative effect as some pollutants build up in the atmosphere over time from continuous emissions. While $CO_2$'s effect is always that of warming, and grows over time from both growing emissions as well as accumulating concentrations due to $CO_2$'s long atmospheric lifetime, the inclusion of all climate pollutants yields a net cooling effect in the near-term consistent with the net negative forcings discussed above. It isn't until the 2080s that shipping's net effect shifts to warming. This is consistent with Unger et al. (2010), who show strong near-term cooling tendencies from the shipping sector that lessen over time as $CO_2$ builds up in the atmosphere. However, note that their study analyzed perpetual year-2000 emissions and not a BAU scenario. This is also consistent with Fuglestvedt et al. (2009), which predicts that the accepted regulations in the shipping sector's emissions of sulfur dioxide and nitrogen oxides will lead to the sector having a net cooling effect for about 70 years, after which the sector switches to warming. Our analysis predicts a slightly more rapid shift to warming (after about 65 years in 2085), likely due to our inclusion of the warming climate pollutant black carbon which are not featured in the analysis by Fuglestvedt et al. (2009).

Based on our BAU projections, future $CO_2$ emissions from international shipping result in an additional warming of 0.07 °C by year 2100. However, when all pollutants are considered, the net warming from shipping in 2100 drops to 0.01 °C due to the net warming and cooling effects from non-$CO_2$ pollutants (Figure 4a).

For the year 2100, the temperature impacts attributed individually to emissions of $CO_2$, black carbon, methane, nitrogen oxides, sulfur dioxide, organic carbon, and carbon monoxide are shown in Figure 4b. The indirect effects of aerosols are included in the analysis of the temperature impacts for each isolated pollutant. Specifically, shipping's cooling effect, which offsets $CO_2$'s warming effect, is dominated by the cooling pollutant precursor nitrogen oxides. The net cooling from nitrogen oxides arises from nitrate formation, indirect aerosol effects from nitrates, formation of tropospheric ozone, reduction of methane, and effects of the net forcings on the carbon cycle (cooling in the ocean suppresses $CO_2$ diffusion from the ocean into the atmosphere). Given that sulfur dioxide emissions—a precursor to the cooling pollutant sulfate—are projected to decrease significantly due to the sulfur fuel regulation newly adopted by IMO, sulfur dioxide from shipping contributes less significantly to cooling. Recent studies have demonstrated the potential for low-sulfur shipping scenarios to reduce the indirect aerosol effect from shipping sulfur emissions (Lauer et al., 2009; Righi et al. 2011). However, the remaining emissions of sulfur dioxide from the shipping sector throughout the century are still responsible for about 0.02 °C cooling by year 2100.

While BAU organic carbon, carbon monoxide, and methane have nearly negligible contributions to shipping's influence on end of century temperatures, shipping's black carbon emissions are responsible for 0.01 °C warming and add to $CO_2$'s warming effects (Figure 4b). We note that for both sectors, our calculations assume no change in the geographical distribution of emissions. Recent literature has demonstrated that the location of non-$CO_2$ emissions can

have a large influence on their subsequent climate impact (Fuglestvedt et al. 2014; Kohler et al. 2013; Fromming et al. 2012; Lund et al. 2017; Skowron et al. 2015).

Figure 5a shows the impact of future international aviation emissions (beginning in 2020) on surface air temperature change throughout the 21$^{st}$ Century. The contribution of $CO_2$ emissions to future warming over time and in the year 2100 is slightly lower than that from the shipping sector (0.05 °C by 2100). However, the inclusion of non-$CO_2$ climate pollutant emissions does not yield a net cooling effect for several decades as they do with shipping, and reduces warming by end of century to 0.03 °C (note that we do not include here the impacts on contrails and cirrus clouds). For a few years, the net temperature impact from future aviation emissions is cooling, but then quickly switches to warming and increases steadily through the end of the century (consistent with radiative forcing calculations in Unger et al. (2010) for constant year-2000 emissions). Similar to shipping, the cooling effect is dominated by the cooling precursor gas nitrogen oxide (Figure 5b). By 2100, nitrogen oxides are responsible for a cooling of 0.02 °C, while the end of century contribution from all other non-$CO_2$ climate pollutants (sulfur dioxide, organic carbon, carbon monoxide, and black carbon) are negligible. Recall that the indirect effects of aerosols are included in the analysis of the temperature impacts for each isolated pollutant. We note that some studies have investigated the effect of aviation soot on natural cirrus clouds (Penner et al., 2019) or the effect on warm clouds (Gettelman and Chen, 2013; Righi et al., 2013; Kapadia et al., 2016). MAGICC does take into account indirect effects of soot, such as simplified parameterizations of impacts on cloud brightness and lifetime, but does not include more sophisticated treatments as analyzed in previous studies.

Our projections for the contribution to future warming from international shipping and aviation are slightly lower than the estimated range for each sector's 2100 share of warming estimated in Skeie et al. (2009). Our estimate of international aviation's contribution to warming is below the 0.11 °C to 0.28 °C range, but Skeie et al. (2009) analyzed combined domestic and international transport emissions. Further, the new emissions projections generated by ICAO in 2019 suggest lower projected emissions from aviation over the next century (ICAO 2019). Our shipping warming impact estimates are at the lower end of the -0.01 °C to 0.25 °C range, attributed to differences in methodology discussed below.

First, our model includes indirect aerosol effects, particularly the climate impact associated with nitrogen oxides' production of nitrate aerosols, which yield negative forcings that are not considered in the analysis by Skeie et al. (2009). This inclusion also explains why nitrogen oxides yield net cooling impacts in our analysis, while they yield net warming impacts by Skeie et al. (2009), due mainly to warming from the production of tropospheric ozone not canceled out by cooling from indirect effects.

Second, our shipping estimates are also lower because Skeie et al. (2009) only consider the emissions of $CO_2$, nitrogen oxides, and sulfur dioxide for each sector, and emissions profiles are based on older projections. In particular, the projected emissions of $CO_2$ and nitrogen oxides in Skeie et al. (2009) are both higher than our projected emissions (which both yield more warming impacts in the absence of indirect aerosol effects), while their emissions of sulfur dioxide are lower than our projections (which means less cooling from sulfate). Acknowledging these differences in

methodology, we observe the same general warming trends within our scenarios and the literature, where aviation emissions exhibit an increasing net warming effect, while shipping emissions result in a declining cooling trend until the end of the century.

In the RCP scenarios presented by Lund et al. (2012), shipping is projected to cause a cooling of between -0.02 and -0.04 °C by midcentury. Our analysis estimates that shipping is responsible for -0.03 °C in year 2050, which falls within this range. Further, the authors' findings are in agreement with those presented in this analysis through their observation of warming later in the century once the accumulating $CO_2$ emissions impact overruns the cooling impact of nitrous oxides and sulfur dioxide, particularly due to the reduced sulfur dioxide emissions associated with the implemented fuel regulations.

Our estimates for the contribution to global average temperature in year 2100 from the aviation sector's $CO_2$ emissions of 0.05 °C falls at the lower end of the range presented by Terrenoire et al. (2019), between 0.04 °C and 0.1 °C, based on a set of eight $CO_2$ emissions projections contrasting in traffic growth and efficiency gains. We note that this analysis includes the impact of both domestic and international aviation. Our estimate of the impact of the aviation sector is also less than that of Huszar et al. (2013), at 0.2 °C or 0.1 °C with and without the impact of the non-$CO_2$ signal, respectively. This analysis also does not account for aviation-produced aerosols and does include the impact of water vapor emissions (as well as that of contrail-cirrus), leading to an elevated warming associated with the sector in comparison to our analysis.

Our model does not include radiative effects from linear contrails nor contrail induced cirrus cloudiness. Although studies suggest a low level of scientific understanding for climate impacts of linear contrails and a very low level of scientific understanding of induced cirrus cloudiness (Lee et al. 2009), considerable work has been made recently towards improving our understanding of these effects. Estimates of the present-day radiative impact of linear contrails range from +3 to +12 mW m$^{-2}$ (Lee et al. 2009; Brasseur et al. 2016), and of cirrus cloudiness range from +12 to +63 mW m$^{-2}$ (Lee et al. 2009; Schumann et al. 2015; Brasseur et al. 2016; Bock and Burkhardt 2016); for context, this is compared to around 30 mW m$^{-2}$ from $CO_2$ emissions – note these values are for both domestic and international aviation. As air traffic rates increase, we expect the radiative forcings from contrails and changes in cirrus cloudiness to increase as well; Bock and Burkhardt (2019) suggest an increase in contrail cirrus radiative forcing by a factor of three from present-day through 2050, due to increases in air traffic and also a slight shift towards higher altitudes.

Without growth in air traffic, inclusion of these effects would increase our radiative forcing estimates in 2100 by 15 to 75% based on the lower and upper estimates of both linear contrails and cirrus cloudiness. Assuming a fivefold growth in air traffic from 2005 to 2100, our radiative forcing estimate from international aviation could increase by 75 to 350%. The resulting impact on temperature responses to BAU international aviation could therefore be considerably higher than our projection of 0.05 °C in 2100: 0.06 to 0.09 °C based on current air traffic patterns and 0.09 to 0.23 °C for a fivefold increase in air traffic.

### 3.2 Avoided warming from mitigation measures

The policy scenarios analyzed have significant potential to reduce the future temperature impacts associated with emissions from the international shipping and aviation sectors (Figure 6). The IMO greenhouse gas target of a 50% reduction in $CO_2$ emissions below 2008 levels by 2050 and full decarbonization of the industry by 2100 results in an

avoided future warming of 0.06 °C by 2100. This avoided warming reduces the shipping sector's contribution to future warming from $CO_2$ by almost 85% at the end of the century. A more stringent mitigation scenario in which decarbonization is achieved by midcentury (consistent with a 1.5 °C warming cap) increases avoided warming to 0.07 °C by 2100, or almost 100% of the original unabated contribution to warming from the sector's $CO_2$ emissions.

Because the non-$CO_2$ climate pollutants emitted by the shipping sector yield a net cooling, the scenarios that reduce

their emissions proportional to the reductions in $CO_2$ outlined in each policy increase each scenario's contribution to future warming and consequently reduce their relative avoided warming. Specifically, the IMO minimum and maximum ambition greenhouse gas targets reduce the anticipated BAU warming from the shipping sector by about 0.02 °C and 0.01 °C by the end of the century, respectively (in comparison to 0.06 °C and 0.07 °C in the $CO_2$-only scenarios, respectively). We expect that the true warming mitigation provided by these policies lies within these

bounds.

The various mitigation scenarios outlined in Table 1 for the international aviation sector result in an avoided future warming of 0.01 °C to 0.05 °C by 2100, relative to a BAU baseline. Full implementation of CORSIA under current guidelines (ending in 2035 and then allowing emissions to increase along a business-as-usual pathway) results in an avoided warming of 0.01 °C by 2100 (a 20% reduction of warming from a $CO_2$ BAU baseline).  However, extending

CORSIA's offsetting and reduction program through the end of the century more than doubles the climate benefit (0.02 °C avoided warming), avoiding about 40% of the $CO_2$ BAU baseline warming. The scenario that follows CORSIA and then decarbonizes the sector by year 2100 (CORSIA-DECARB2100) reduces future warming by 0.04 °C by end of century, avoiding about 80% of the $CO_2$ BAU warming. The most aggressive mitigation policy, completing CORSIA followed by decarbonization of the sector by 2050, results in an avoided warming of 0.05 °C by

2100 (about a 90% reduction of warming from a $CO_2$ BAU baseline). The avoided warming in year 2100 associated with each investigated policy scenario for the emissions mitigation of international shipping and aviation are outlined in Figure 7.

The warming mitigation potential of the various policy scenarios associated with aviation were evaluated based on three $CO_2$ BAU growth patterns. While the BAU pathway dictates the magnitude of projected future warming, the

associated avoided warming from each policy scenario is relative to the BAU baseline.  For international aviation, in comparison to the 0.05 °C contribution to future warming from $CO_2$-only expected from the central growth rate pattern, 0.10 °C and 0.02 °C of future warming are expected from the emission of $CO_2$ in the upper and lower emissions growth rate patterns, respectively. The mitigation scenario that mimics the structure of the IMO minimum ambition greenhouse gas target (CORSIA – DECARB2100), for example, is expected to reduce the warming attributed to the

emissions of $CO_2$ from the sector by 0.04 °C by end of century in the limited growth rate pattern, and is expected to

avoid 0.09 °C and 0.01 °C by end of century in the upper and lower growth rate patterns, respectively. These avoided temperatures from the upper, central, and lower growth scenarios represent 87%, 84%, and 66% of the unabated warming levels from $CO_2$ emissions, respectively. While the expected BAU warming from the sector's $CO_2$ emissions vary significantly between each pattern of growth, the potential to reduce this warming through proposed, stringent

mitigation scenarios scales proportionally for the two higher emissions growth rate scenarios. In contrast, the BAU lower growth scenario demonstrates a future in which emissions remain relatively close to 2020 levels throughout the century. Because the most stringent policy scenarios investigated in this analysis focus on emissions reductions taking place in mid-to late-century, a lower percent warming reduction is observed for each policy within the lower growth scenario in comparison to the upper and central scenarios.

Although we do not expect that the offsetting programs analyzed here will affect the amount of contrail and cirrus cloud formation, and therefore will not impact the avoided warming potential, improvements to aircraft technology and management practices may reduce the prevalence and thickness of these clouds, for example due to increased fuel efficiency. Similarly, we do not consider the reduction of non-$CO_2$ climate pollutants emitted by the aviation sector in the mitigation scenarios. However, offsetting schemes such as CORSIA do implement the use of biofuels and

aircraft technology and air traffic management improvements, both of which have the potential to impact future emissions of non-$CO_2$ climate pollutants and the density of contrail cirrus (Bock and Burkhardt 2019; Caiazzo et al., 2017; Burkhardt et al., 2018). While the influence of these changes on the non-$CO_2$ impact of international aviation is currently not well-estimated, their impact should be considered in future analyses as understanding develops.

**4 Conclusions**

Quantifying the temperature impacts of future international aviation and shipping emissions—both for business-as-usual pathways and mitigation scenarios—is essential to understanding the benefits of proposed policies and targets. Given that international aviation and shipping are important contributors to the emission of climate pollutants, earlier studies have analyzed their current and BAU future climate impacts using a variety of methods. To build upon these previous analyses, we analyzed the climate benefits over time associated with accepted, proposed, and prospective

mitigation policies for each sector. We use a reduced complexity climate model to determine the BAU temperature contribution due to the future emissions of international aviation and shipping from all emitted climate pollutants, and the potential to avoid future warming based on a series of realistic mitigation scenarios.

Using the reduced complexity climate model MAGICC, we estimate that under BAU conditions, the future $CO_2$ emissions (2020 through end of century) from the international shipping and aviation sectors would be responsible for

0.07 °C and 0.05 °C of future warming by 2100, respectively (0.01 °C and 0.03 °C, respectively, when including the sectors' emissions of non-$CO_2$ climate pollutants; additional inclusion of aviation induced contrails and clouds could increase the warming associated with international aviation by up to 600%). Planned and proposed mitigation policies in each sector that specifically target $CO_2$ emissions have the potential to significantly reduce this climate impact. However, policies that target the mitigation of non-$CO_2$ climate pollutants, often through air quality management,

result in emissions reductions that may not always avoid future warming (Kapadia et al., 2016; Sofiev et al., 2018;

Yim et al., 2015). For example, if the emissions of all shipping-produced cooling agents (sulfur dioxide, nitrogen oxides, and organic carbon) were immediately halted and the shipping sector successfully decarbonized by midcentury, the sector would increase the world's temperatures through the end of the century (Fuglestvedt et al. 2009).

Given that we have already reached a global warming level of around 1 °C above preindustrial levels (IPCC, 2018), there is an "allowable warming" of 0.5 to 1.0 °C additional warming should we wish to stabilize at the 1.5 °C or 2 °C thresholds, respectively. Together, future warming from the $CO_2$ emissions of international shipping and aviation reach about 0.12 °C by the end of the century, which is 12-24% of this remaining "allowable warming." However, certain policy measures have the potential to significantly avoid the vast majority of this future warming. The IMO
minimum ambition greenhouse gas target (decarbonize the sector by 2100) and its mirrored aviation scenario (CORSIA offsetting program extended followed by decarbonizing by 2100) have the potential to reduce future warming associated with the $CO_2$ emissions from each sector by more than 80%, with even further reductions should both sectors decarbonize by midcentury in comparison to 2100 (a trajectory consistent with achieving 1.5 °C maximum warming).

For context, achieving the Paris Agreement committed pledges and targets are projected to avoid 0.3 °C warming by end of century compared to current policies (CAT, 2019). Adding the avoided warming from the already agreed upon international shipping target of decarbonization by end of century (0.06 °C) and the extension of the CORSIA aviation offsetting program (0.02 °C) increases this potential by over 25%. Further, pursuing the most ambitious, yet feasible, mitigation measures for international shipping and aviation could increase the avoided warming from the Paris
Agreement by nearly 50%. Overall, the proposed and prospective mitigation measures for both of these sectors have considerable climate benefits in the context of achieving international temperature goals.

**Code availability**

The MAGICC v6 model executable is available for download at: http://www.magicc.org/download upon registration, although the model itself is closed source. The user manual can be accessed at:
http://wiki.magicc.org/index.php?title=Manual_MAGICC6_Executable. Full model details along with nineteen sets of AOGCM-calibrated parameters used here for ensemble members are found in Meinshausen et al. (2011a). We update the default values of methane and tropospheric ozone radiative efficiency and methane atmospheric lifetime to values in Myhre et al. (2013) and Etminan et al. (2016).

**Data availability**
Results from the MAGICC model are available from Catherine Ivanovich (civanovich@edf.org) upon request.

**Author contributions**

Catherine Ivanovich and Ilissa Ocko designed the experiments and Catherine Ivanovich carried them out. Annie Petsonk and Pedro Piris-Cabezas curated data and provided guidance on the policies. Catherine Ivanovich and Ilissa Ocko prepared the manuscript with contributions from all co-authors.

**Acknowledgements**

Catherine C. Ivanovich was funded by the High Meadows Foundation. Ilissa B. Ocko was funded by the Heising Simons Foundation and the Robertson Foundation. We thank Nathaniel Keohane, and Steven Hamburg for reviewing our manuscript.

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

| | Mitigation scenario description | Abbreviation |
|---|---|---|
| **Aviation** | Cap emissions at 2020 levels | CAP |
| | CORSIA (including offsets, biofuel use, and improvements to aircraft technology and air traffic management) ends after 2035, followed by business-as-usual emissions growth | CORSIA |
| | CORSIA emissions reductions sustained through 2100 | CORSIA – EXT |
| | CORSIA ends in 2035, followed by decarbonization in 2100 | CORSIA – DECARB2100 |
| | CORSIA ends in 2035, followed by decarbonization in 2050 | CORSIA – DECARB2050 |
| **Shipping** | IMO Greenhouse Gas Targets: 50% reduction from 2008 levels by 2050, decarbonization by 2100; does not affect non-$CO_2$ pollutants | IMO – MIN AMBITION, $CO_2$ ONLY |
| | Linear decrease in emissions starting in 2020, leading to decarbonization in 2050; does not affect non-$CO_2$ pollutants | IMO – MAX AMBITION, $CO_2$ ONLY |
| | IMO Greenhouse Gas Targets: 50% reduction from 2008 levels by 2050, decarbonization by 2100; proportional emissions reductions for all non-$CO_2$ pollutants | IMO – MIN AMBITION, ALL POLLUTANTS |
| | Linear decrease in emissions starting in 2020, leading to decarbonization in 2050; proportional emissions reductions for all non-$CO_2$ pollutants | IMO – MAX AMBITION, ALL POLLUTANTS |

**Table 1: Descriptions of mitigation scenarios analyzed in this study for international aviation and shipping.**

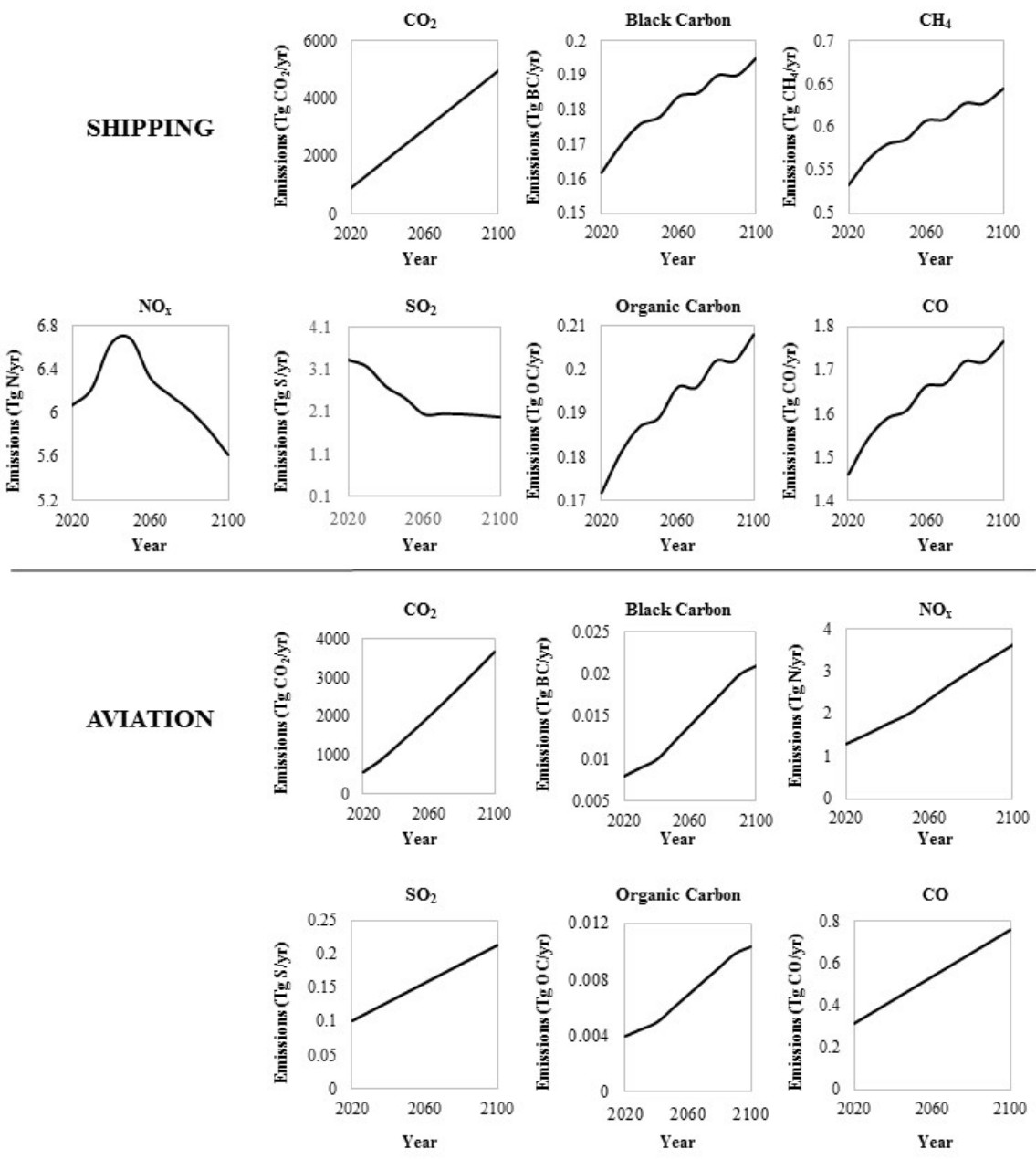

**Figure 1: Projected future emissions from international shipping and aviation. Shipping $CO_2$ emissions data from Third IMO Greenhouse Gas Study (IMO 2014) and the Update of Maritime Greenhouse Gas Emission Projections (Hoen et al. 2017). Aviation $CO_2$ emissions data from Present and Future Trends in Aircraft Noise and Emissions (ICAO, 2019). Both datasets end in 2050; shipping data is linearly extrapolated through year 2100 and aviation data utilizes the described growth extrapolation after 2040 through year 2100. Aviation black carbon and $NO_x$ emissions and shipping black carbon, $CH_4$, $NO_x$, $SO_2$ (adjusted based on IMO's recently adopted sulfur fuel regulation), organic carbon, and CO extracted from RCP Database for the RCP8.5 scenario. Aviation $SO_2$ and CO are linearly extrapolated from the EDGAR dataset, and aviation organic carbon emissions are derived from their relationship with black carbon emissions.**

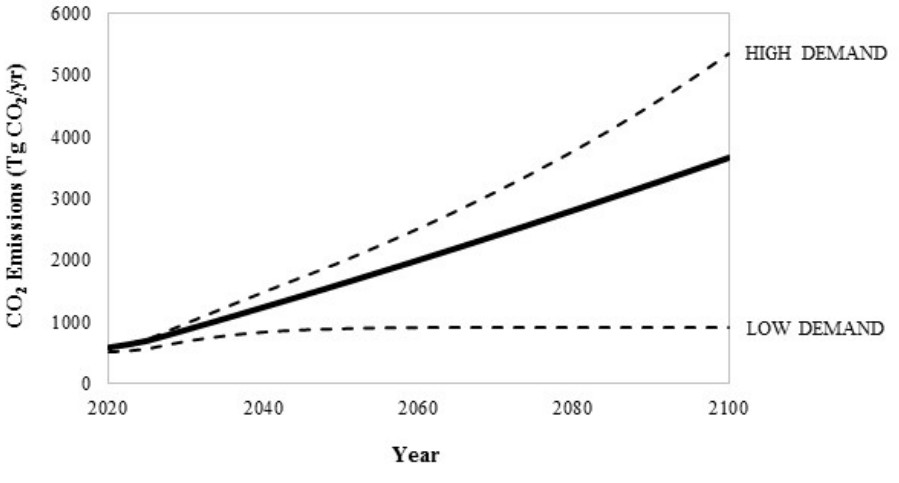

**Figure 2: Projected future emissions from international aviation used for sensitivity analysis. The sensitivity tests (dashed lines) are based on an exponential growth rate pattern through 2100 following the 2005-2050 trend for the high and low demand forecasts as depicted in a previous version of Present and Future Trends in Aircraft Noise and Emissions (ICAO, 2013).**

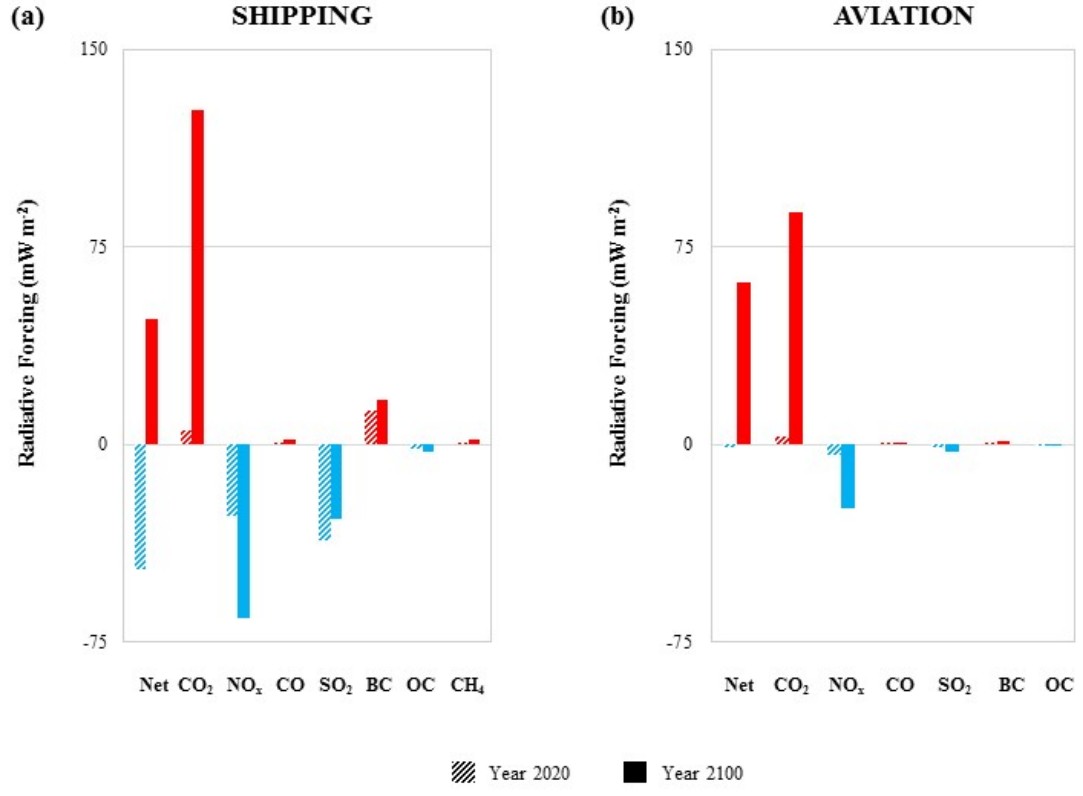

**Figure 3: Contribution of emittants to radiative forcing in 2100 (defined as the forcing at the tropopause after stratospheric temperature adjustment) from business-as-usual emissions from a) international shipping and b) international aviation. Radiative forcings are presented for year 2020 (hashed), which represent forcings from emissions that year, and year 2100 (solid), which represent the change in forcings from 2020 to 2100.**

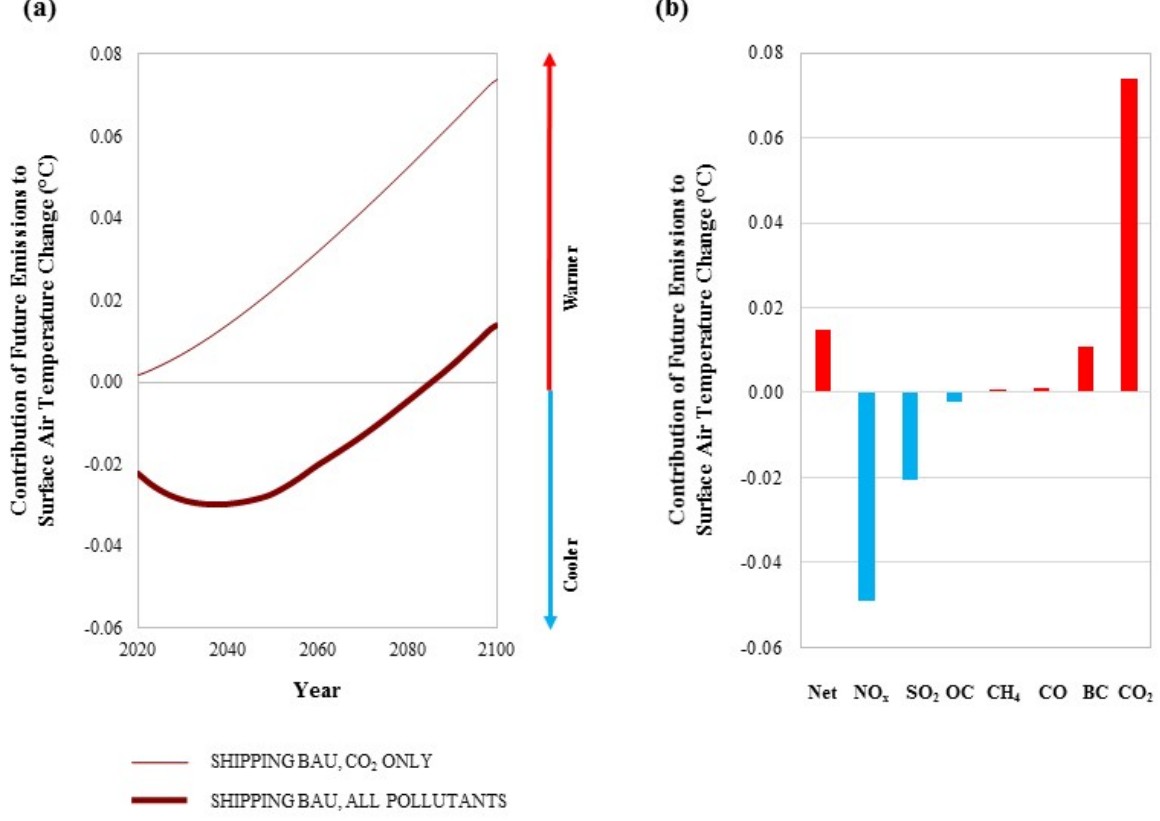

**Figure 4: Contribution of future emissions to surface air temperature change in °C associated with business-as-usual emissions starting in 2020 and continuing through the end of the century from international shipping. Future temperature impacts are presented a) for emissions of $CO_2$ only (thin line) and all pollutants (thick line), and b) by contribution of individual pollutants in year 2100.**

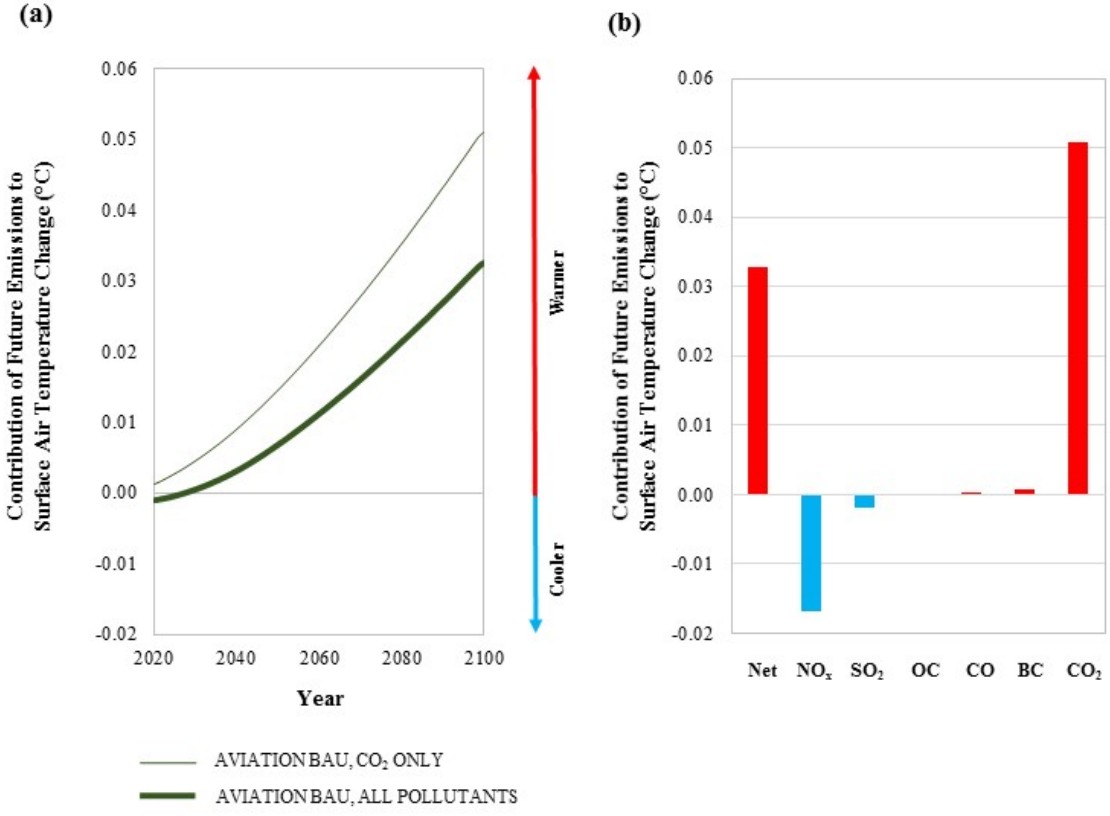

**Figure 5: Contribution of future emissions to surface air temperature change in °C associated with business-as-usual emissions starting in 2020 and continuing through the end of the century from international aviation. Future temperature impacts are presented a) for emissions of CO₂ only (thin line) and all pollutants (thick line), and b) by the contribution of individual pollutants in 2100.**

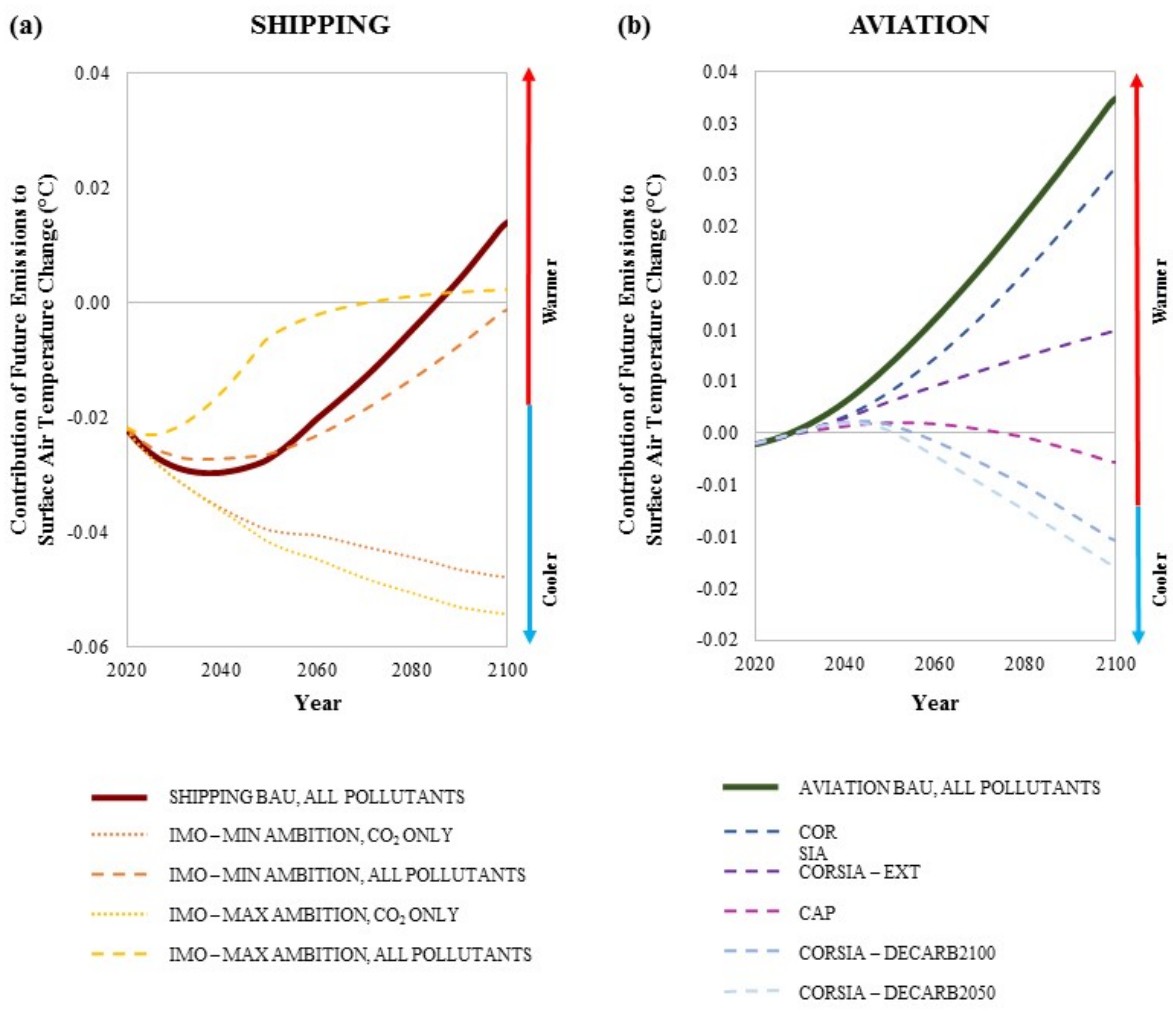

**Figure 6: Surface air temperature changes associated with various policy scenarios for emissions mitigation in international a) shipping and b) aviation. Each business-as-usual scenario presents the contribution to future surface air temperature from the emissions of all climate pollutants starting in 2020 and continuing through the end of the century.**

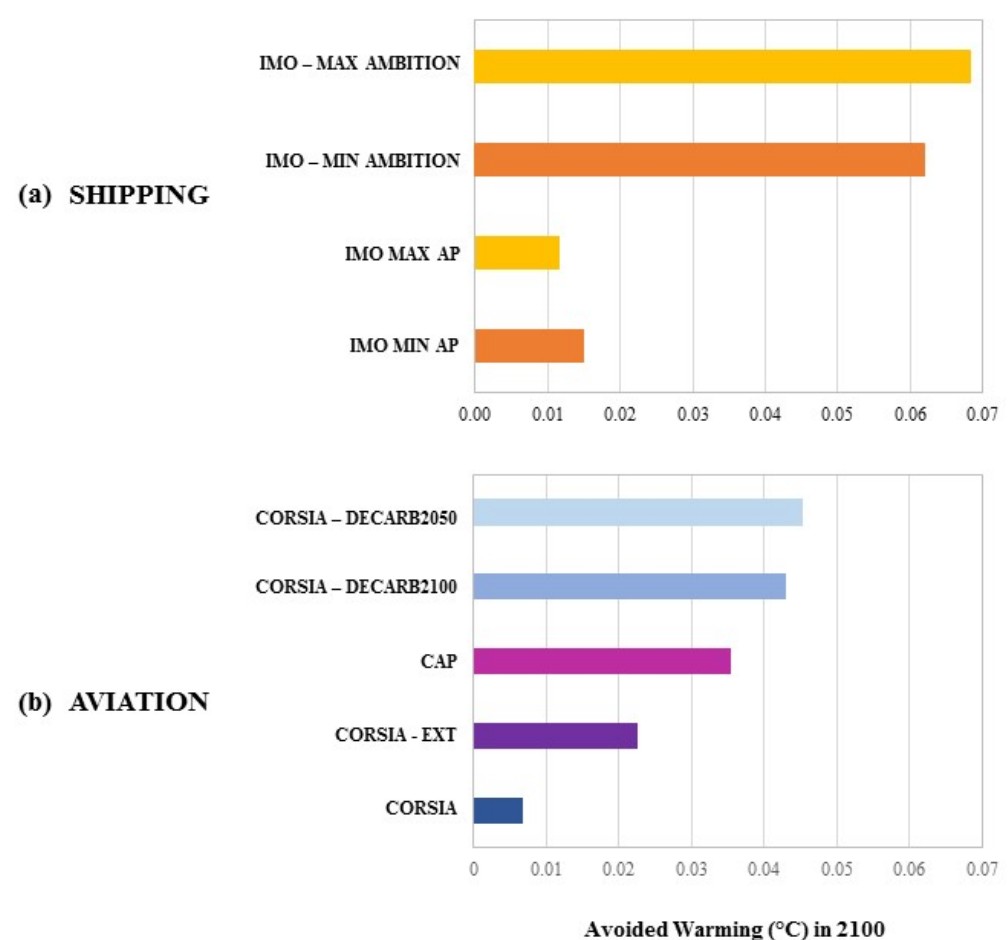

**Figure 7: Avoided warming in year 2100 associated with various policy scenarios for emissions mitigation in international a) shipping and b) aviation.**