# Peer review of "Climate benefits of proposed carbon dioxide mitigation strategies for international shipping and aviation"

_Atmospheric Chemistry and Physics, 2019_

## Referee Comment (RC1) · Anonymous Referee #2 · 3 May 2019

In this study, a simplified climate model is applied to analyze the impact of several $CO_2$ mitigation scenarios for the international shipping and aviation sectors. Not only the impact of $CO_2$, but also of other co-emitted short-lived compounds is considered and the resulting effects on near surface air temperature up to 2100 are quantified.

The study focuses on a very important topic in view of the 1.5 degree goal set by the Paris Agreement and the ways to achieve it. It is well structured, clearly written, and in my opinion fits well to the scope of ACP.

A major criticism is that the results section is quite short and mentions very few previous studies. I would recommend to extend Section 3, adding more details and more

citations, in particular concerning the role of the short-lived species. These compounds can be very relevant for the two sectors discussed here, as shown by several previous studies (see suggestions below). Also the uncertainties of the adopted simplified climate model in simulating the effects of short-lived species can be large and should be discussed.

I find nevertheless the manuscript suitable for publication, after addressing the detailed comments and suggestions listed below.

MAJOR COMMENTS:

- in Sect. 2.1, scaling shipping $SO_2$ emissions by a factor 7 to account for the IMO regulations in fuel sulfur content (FSC) only makes sense if the RCP8.5 dataset assumes a 3.5% FSC for the global shipping fleet. Is this really the case? The 3.5% cap was enforced in 2012, but it was 4.5% before and the RCPs scenarios start the projection in 2000. Moreover, according to the second IMO Study (Buhaug et al., 2009), the actual FSC in the global shipping fleet was on average 2.7% before the introduction of the IMO regulations. Therefore it could be that the FSC value assumed in RCP8.5 is lower than 3.5% and the scaling factor to get to 0.5% is lower than 7. Please check this.

- end of Sect. 2.3: I understand that a full discussion of the model uncertainties is beyond the scope of this study, but I would at least briefly summarize which of them are the most significant for the results presented here.

- P8, L2: you may also want to compare with Lund et al. (Environ. Sci. Technol., 2012).

- P8, L16: there are a few studies simulating the aerosol indirect effect in low-sulfur shipping scenarios you may want to mention, for example Lauer et al. (Environ. Sci. Technol., 2009) and Righi et al. (Environ. Sci. Technol., 2011).

- P8, L26-27: the switch from cooling to warming is not evident in Fig. 2b. Does it occur before 2020? Please clarify.

- P8, L28-33: the issue of aviation effects of short-lived species should be discussed in

more detail (see Lee et al., Atmos. Environ., 2010; or Grewe et al., Aerospace, 2018). There are several studies arguing for the effect of aviation soot on natural cirrus clouds (e.g., Penner et al., J. Geophys. Res., 2019) and some groups even argued for an effect on warm clouds (Gettelman and Chen, Geophys. Res. Lett., 2013; Righi et al., Atmos. Chem. Phys., 2013; Kapadia et al., Atmos. Chem. Phys., 2016). Can the simple climate model used here account for these effects?

- P8, L29: what is the mechanism behind the cooling effect from nitrogen oxide? This gas can lead to the formation of ozone, which has a warming effect, but it also reduces methane lifetime, resulting in a cooling. Are these mechanisms included in the model?

- P8, L31-33: this is confusing, if you include the indirect aerosol effects, then you do address the impacts of aviation on cloudiness. You probably mean contrails and contrail-induced cloudiness here; please clarify and also add a citation to support the last statement in this sentence (warming effect).

- end of Sect. 3.1: only one study is cited for comparison. It would be good to add more, possibly more recent, studies.

- Sect. 3.2: the role of short-lived pollutants in the aviation scenarios is not discussed at all. I understand that, unlike the shipping scenarios, the analyzed aviation scenarios do not distinguish between $CO_2$ and non-$CO_2$ species, but at least some qualitative considerations could be added here.

MINOR SUGGESTIONS / CORRECTIONS:

- P1, L14: please specify how much is this allowable warming.

- P1, L33: how does the time-frame affect the share of global $CO_2$ emissions? Shouldn't it be rather given for a specific year?

- P3, L18: I would change the title of Sect. 2.1, to make more clear that the baseline scenario is discussed here. I would also suggest to make two subsections of 2.1, to better separate aviation from shipping. The same would apply to 2.2. Another option

would be to merge 2.1 and 2.2 in a single section on emissions, with two subsections for shipping and aviation, respectively.

- P3, L27: "hold that level constant", I guess you are referring to the growth rate which is held constant, but that could be misunderstood as the actual emissions. I would be more explicit: "hold that growth rate constant".

- P4, L5: the RCP acronym should be explained.

- P4, L11: what do you mean by "all-forcings" BAU scenario?

- P4, L29: there are more recent estimates, for example Burkhardt and Kärcher (Nature Clim. Change, 2011).

- P6, L24: please provide a reference for these relationships.

- P6, L26: please replace "gas" by "species" or "compound", since also aerosols are considered here.

- P7, L4: 2100 - 1765 + 1 = 336 years (?)

- P11, L13: it might be worthwhile to cite Fuglestvedt et al. (Environ. Sci. Technol., 2009) in this context.

- P11, L14-16: since this is the main motivation behind this work, I would suggest putting this sentence also in the introduction.

- Figure 1: the acronym MMT should be explained. Also, this is a non-SI unit: I would use Tg or Gg instead.

- Figure 3a is discussed before Figure 2b. You could think about grouping the plots by sector since this reflects the way they are presented in the text.

- Figure 4: it is hard to distinguish the lines for the different scenarios, since very similar colors are used for them.

---

## Referee Comment (RC2) · Anonymous Referee #1 · 7 May 2019

This paper investigates the potential for reducing the climate impact of international aviation and shipping over the 21st century. The policies and targets adopted by the IMO and ICAO are translated into emissions scenarios and the subsequent impact on global mean temperature response assessed using the reduced complexity climate model. Knowledge of how these sectors can reduce the climate footprint and align with the goals of the Paris Agreement is important and the study hence provide a topical contribution. The paper is well-written and clearly structured. I do, however, have some concern about the methodological assumptions, and would like to see a better treatment of uncertainties as well as better integration with the current body of literature and status of knowledge before the paper can be published in ACP.

[Figure]

General comments: I believe that excluding the contrail-cirrus effect is a major limitation of this study, which, although the authors repeat the caveat, gives misleading results. I find it particularly problematic since the authors choose to include indirect effects due to aerosols, which are much less studied and have even larger uncertainties. Moreover, the authors refer to a paper by Skeie et al. 2009, and in this study an estimate of the contrail impact was included. At the very least, the authors need to better justify this choice and better reflect the large body of literature on contrail-cirrus to give a qualitative estimate of how their results could look if this was included. Some modification of how the temperature impacts of all pollutants are presented should be made.

Another major limitation is the lack of uncertainty assessment. I strongly encourage the authors to include some sort of uncertainty calculation here, given the large uncertainties in RF estimates and climate sensitivity.

Related to this, I also feel that the discussion is incomplete and that there are several issues where further elaboration and caveats are needed, and where existing literature should be better reflected. Moreover, additional results such as emission pathways and RF estimates should be provided in order for the reader to better be able to understand the drivers of the temperature response, potential impact of uncertainties and differences compared to previous results. See specific comments below.

Specific comments:

Pg1, line 27: "emissions from these" – should this be "emission reductions from these"?

Pg1, line 34-35: What is meant by "over a 20- and 100-year timeframe"? Is this because you're talking about $CO_2$-equivalent emissions? Please correct/clarify.

Pg3, line 31-35: does this mean that there are in fact four BAU scenarios for aviation, the three sensitivity ones and the one described above on lines 27-30? Please clarify.

Pg3, line 31-35: it would be very helpful for the reader if the future emission pathways

under the scenarios and alternative BAU were shown.

Pg4, line 4: what's the rationale for selecting this one?

Pg4, line 29-30: I think this sentence fails to take into account the large amount of previous and ongoing work on contrail-cirrus across many groups. While certainly true that there is an significant uncertainty bar on the contrail-cirrus RF estimate, significant progress has been made over recent years and I encourage the authors to reflect this.

Pg. 4, line 26: What about the even more uncertain indirect effect of shipping sulfate aerosols? Is that included and how?

Pg4, line 32: this is not necessarily the case if the offsetting schemes include a switch to biofuels – see e.g., Caiazzo et al. 2017 ERL, Burkhardt et al. 2018 npj Climate and atmospheric science.

Pg6, line 9: what is the climate sensitivity of MAGICC?

Pg6, line 24: please be more specific. Are particular parameterizations for the aviation and shipping sectors used? Also, given the large uncertainties in the RF of many climate-relevant components, which in turn are critical for the total temperature impact (see also comment below) and hence the contribution of aviation and shipping, the authors need to provide information about the RF estimates (present day relative to pre-industrial) underlying their simulations. In particular, RF estimates specific to aviation and shipping – e.g., what is the indirect aerosol effect of shipping and aviation? And NOx-induced O3 and CH4 effect of aviation? This will allow the reader to better compare to previous literature and assess differences between studies. The current study should be compared with previous results. While, as the authors state early on, there are limited number of studies of temperature impact of different sectors, there is a large body of literature on RF.

Pg7, line 14 – onwards: in this study, I believe all aviation scenarios are compared with the same baseline for global emissions (RCP8.5)? I think a nice addition would be to

discuss the sectors contributions given that not only they, but also the rest of the world makes progress on emissions.

Pg7, line 30: again, this is an example of where information about underlying RF is critical and should be compared with previous literature.

Pg8, line 2: compare with other studies? E.g., Fuglestvedt et al. 2009.

Pg8, line 19-21: for both sectors, the authors should also note that the calculations assume no change in geographical distribution of emissions. For non-CO2 emissions, location can be critical for the subsequent impact. E.g., Fuglestvedt et al. 2014; ES&T, Köhler et al. 2013 Atm. Environ; Frömming et al. 2012 JGR; Lund et al. 2017 ESD

Pg8, line22 – onwards: As already pointed out in the major comment, I believe this result in misleading given the lack of treatment of contrail-cirrus. While it possible that the indirect aerosol effects of aviation sulfate and BC could be negative enough to cause a net cooling, there is nothing in our current best understanding that suggests so. If included I think the authors should make a point of the missing effects at the very start of this paragraph not at the bottom, emphasizing that one should be careful not to read too much into this finding.

Pg8. line 31: again, the estimates of indirect aerosol RF should be included for comparison with e.g., Gettleman et al. 2013 GRL.

Pg9, lines 7-10: Skeie et al. 2009 included both indirect aerosol effects and contrail-cirrus forcing – see their figure 2. Please correct or specify which indirect effects beyond there is included in this analysis.

Pg9, line 9: is the net NOx RF negative in Skeie et al. for shipping?

Pg11, line 4: the IPCC report on 1.5 degrees showed that there was a large difference between temperature response and time until reaching temperature thresholds between two simplified climate models. Uncertainties in the background temperature response affects the contribution from aviation and shipping, and should be discussed

somewhere in the paper (perhaps the authors should consider a dedicated discussion section).

Figure 4: the authors should show also the CO2 only cases here, as in Figure 2, allowing the reader to assess the impact of the assumption that non-CO2 emissions are changed "proportionally".

[Figure]

---

## Author Response (AR1)

**Manuscript Ref: acp-2019-126**

**Climate benefits of proposed carbon dioxide mitigation strategies for international**

**shipping and aviation**

Catherine C. Ivanovich, Ilissa B. Ocko, Pedro Piris-Cabezas, and Annie Petsonk

We sincerely appreciate the careful reviews and helpful suggestions provided by the Reviewers, and thank the Reviewers and the Editor for their time. The manuscript has been considerably improved and strengthened based on the changes in response to the comments. Below, we provide information on the major modifications to the paper and respond point-by-point to comments (reviewer comments in blue, responses in black).

Major changes to the paper include:
- The addition of a new figure (Figure 2) providing the emissions profiles associated with the sensitivity analyses for business-as-usual aviation $CO_2$ emissions.
- The addition of a new figure (Figure 3) showing the radiative forcing estimates for each emitted climate pollutant from the international shipping and aviation industries.
- The inclusion of (27) additional references.
- Improved representation of the current status of emissions regulation in the international shipping and aviation sectors and associated reconfiguring of the modeling scenarios.
- The incorporation of updated $CO_2$ emissions projections for international aviation from recently published data from the International Civil Aviation Organization.
- The inclusion of an analysis of the potential increase in warming associated with the aviation sector due to the consideration of contrails and contrail-cirrus.
- Explicit comparison of the BAU radiative forcing estimates with those previously published.

**Responses to Anonymous Referee #1:**

**SPECIFIC COMMENTS**

**Comment 1:** Pg1, line 27: "emissions from these" – should this be "emission reductions from these"?

> **Response:** We appreciate this observation, and we have edited the language accordingly to read, "*The Conference of the Parties to the United Nations Framework Convention on Climate Change (UNFCCC) in the late 1990s urged that emissions reductions from these sectors be pursued through the UN's International Civil Aviation Organization (ICAO, established 1944) and International Maritime Organization (IMO, established 1948), respectively (UNFCCC, 1997)*" on P1:L28.

**Comment 2:** Pg1, line 34-35: What is meant by "over a 20- and 100-year timeframe"? Is this because you're talking about CO2-equivalent emissions? Please correct/clarify.

> **Response:** Given that we were initially calculating international aviation and shipping's share of global *greenhouse gas* emissions, we had to employ $CO_2$-equivalents (which require a time horizon). However, we have since realized that this sentence overcomplicated our message, and we are now comparing these sectors' emissions to energy-related $CO_2$ emissions worldwide – eliminating the need for a time horizon. The text now reads: "*While current emissions from international aviation and shipping account for around 4% of global energy-related $CO_2$ emissions (IMO, 2014; ICAO, 2019a; IEA, 2018), emissions from each sector are forecasted to increase anywhere from 200-400% (Lee 2018) and 50-250% (IMO, 2014) by midcentury, respectively, in the absence of effective policy.*" on P2:L1.

**Comment 3:** Pg3, line 31-35: does this mean that there are in fact four BAU scenarios for aviation, the three sensitivity ones and the one described above on lines 27-30? Please clarify.

> **Response**: There are three total BAU scenarios analyzed for aviation: the central scenario used for the analysis (described on P4:L5) and two associated with sensitivity tests. We have reorganized the section in which these scenarios are described, and provided additional clarification in order to help make this more apparent. The section now reads as follows on P4:L8: "*Given that there is a range of reasonable growth patterns for aviation emissions in particular (Lee, 2018; Skeie et al., 2009), and our results depend on this baseline, we ran a set of sensitivity tests to evaluate the influence of different $CO_2$ BAU projection growth patterns on the perceived avoided warming impacts. The two sensitivity tests considered are based on an exponential growth rate pattern through 2100 following the 2005-2050 trend for the high and low demand forecasts as depicted in a previous version of Present and Future Trends in Aircraft Noise and*

*Emissions (ICAO, 2013). These emissions estimates are scaled down to calculate the corresponding Low Aircraft Technology and Moderate Operational Improvement Scenarios proportionally to the latest ICAO forecast (2019a), resulting in declining growth rate patterns in which growth rates follow their 2020-2050 declining trend until plateauing at 0% –as is the case for the low demand scenario. These sensitivity tests are analyzed in addition to the Low Aircraft Technology and Moderate Operational Improvement Scenario noted above, for a total of three analyzed BAU scenarios for aviation."* We note that the emissions profiles for the central scenario and the two sensitivity tests have changed slightly based on the release of new $CO_2$ emissions projections for international aviation provided by a released working paper from the International Civil Aviation Organization (ICAO, 2019a). The emissions projections are highlighted in the new Figure 2.

**Comment 4:** Pg3, line 31-35: it would be very helpful for the reader if the future emission pathways under the scenarios and alternative BAU were shown.

**Response:** We thank the referee for this thoughtful suggestion. We have created an additional figure (Figure 2, reproduced below) in order to outline the emissions pathways under the central and alternative BAU scenarios associated with the sensitivity analyses for international aviation.

[Figure]

**Comment 5:** Pg4, line 4: what's the rationale for selecting this one?

**Response:** We thank the reviewer for bringing this question to our attention and highlighting that its answer is not provided in the submitted text. We now provide additional context as to why the central growth scenario was chosen for all figures relating to the future warming associated with aviation. Specifically, because the central growth scenario represents the middle of the road scenario, it allows us to

avoid extreme estimations on either side of the spectrum. This explanation is provided on P4:L20, reading as follows: "*We note that all other figures in the paper reflect the Low Aircraft Technology and Moderate Operational Improvement Scenario, which depicts a limited growth pattern for international aviation as this provides a middle of the road estimation.*"

**Comment 6:** Pg4, line 29-30: I think this sentence fails to take into account the large amount of previous and ongoing work on contrail-cirrus across many groups. While certainly true that there is an significant uncertainty bar on the contrail-cirrus RF estimate, significant progress has been made over recent years and I encourage the authors to reflect this.

> **Response:** We agree with the reviewer and have refined the text in the Methods section to acknowledge the recent work towards understanding these impacts: "*The latest version of MAGICC is not calibrated for inclusion of linear contrails and induced cirrus cloudiness from aviation, phenomena in which water vapor and impurities released in aircraft exhaust form cirrus-like clouds. This is an active area of research and significant progress has been made in recent years to better understand these uncertain processes (e.g. Lee et al. 2009; Schumann et al. 2015; Brasseur et al. 2016; Bock and Burkhardt 2016).*" (P7:L1).
>
> We have also added text to address this in the Results section, along with discussion of the estimates in the literature and a sensitivity analysis to show the potential impact on our BAU radiative forcing estimates and temperature responses to aviation. The additional text reads: "*Our model does not include radiative effects from linear contrails nor contrail induced cirrus cloudiness. Although studies suggest a low level of scientific understanding for climate impacts of linear contrails and a very low level of scientific understanding of induced cirrus cloudiness (Lee et al. 2009), considerable work has been made recently towards improving our understanding of these effects. Estimates of the present-day radiative impact of linear contrails range from +3 to +12 mW m$^{-2}$ (Lee et al. 2009; Brasseur et al. 2016), and of cirrus cloudiness range from +12 to +63 mW m$^{-2}$ (Lee et al. 2009; Schumann et al. 2015; Brasseur et al. 2016; Bock and Burkhardt 2016); for context, this is compared to around 30 mW m$^{-2}$ from CO$_2$ emissions – note these values are for both domestic and international aviation. As air traffic rates increase, we expect the radiative forcings from contrails and changes in cirrus cloudiness to increase as well; Bock and Burkhardt (2019) suggest an increase in contrail cirrus radiative forcing by a factor of three from present-day through 2050, due to increases in air traffic and also a slight shift towards higher altitudes.*
>
> *Without growth in air traffic, inclusion of these effects would increase our radiative forcing estimates in 2100 by 15 to 75% based on the lower and upper estimates of both linear contrails and cirrus cloudiness. Assuming a fivefold growth in air traffic from 2005 to 2100, our radiative forcing estimate from international aviation could increase by 75 to 350%. The resulting impact on*

*temperature responses to BAU international aviation could therefore be considerably higher than our projection of 0.05 °C in 2100: 0.06 to 0.09 °C based on current air traffic patterns and 0.09 to 0.23 °C for a fivefold increase in air traffic.*" (P12:L23).

**Comment 7:** Pg. 4, line 26: What about the even more uncertain indirect effect of shipping sulfate aerosols? Is that included and how?

> **Response:** We do include indirect effects from all aerosols. We have added text to acknowledge the uncertainties in both aerosol direct and indirect forcings: "*Whereas radiative impacts of well-mixed greenhouse gases (such as $CO_2$ and methane) are fairly well understood due to our knowledge of gas absorption, aerosol radiative effects are more complex and uncertain. This is due to spatial and temporal heterogeneity complicating observations; a variety of possible microphysical and optical properties based on varying sizes, shapes, structures, mixtures, and humidity levels; and interactions with clouds that can impact the lifetime and brightness of the clouds. Given that aerosols are quite relevant to both the aviation and shipping sectors (e.g. Unger et al., 2010), we include their direct and indirect effects in our simulations, noting that caution must be applied in interpreting the results. Aerosol direct forcings are approximated by simple linear forcing-abundance relationships. The indirect effects of sulfate, black carbon, organic carbon, nitrate, and sea salt aerosols are also included. The effect on cloud droplet size is determined by scaling optical thickness patterns of each species (as described by Hansen et al. (2005)) by their respective emissions. The effect of aerosols on cloud cover and lifetime is modeled as a prescribed change in efficacy of the cloud albedo (for full parameterization details, see Meinshausen et al. (2011a))*" (P6:L17).

**Comment 8:** Pg4, line 32: this is not necessarily the case if the offsetting schemes include a switch to biofuels – see e.g., Caiazzo et al. 2017 ERL, Burkhardt et al. 2018 npj Climate and atmospheric science.

> **Response:** We thank the referee for bringing this point to our attention. We have updated this section to include a description of the potential for offsetting schemes including a switch to biofuels to impact the climate benefit of associated policies. We have also highlighted the suggested citations on P14:L19, reading: "*However, offsetting schemes such as CORSIA do implement the use of biofuels and aircraft technology and air traffic management improvements, both of which have the potential to impact future emissions of non-$CO_2$ climate pollutants and the density of contrail cirrus (Bock and Burkhardt 2019; Caiazzo et al., 2017; Burkhardt et al., 2018).*"

**Comment 9:** Pg6, line 9: what is the climate sensitivity of MAGICC?

**Response:** The equilibrium climate sensitivity of MAGICC is 3 °C, and can be found on P6:L7: "*MAGICC contains a hemispherically averaged upwelling-diffusion ocean coupled to a four-box atmosphere (one over land and one over ocean for each hemisphere) and a carbon cycle model, with an average equilibrium climate sensitivity (ECS) of 3 °C.*"

**Comment 10:** Pg6, line 24: please be more specific. Are particular parameterizations for the aviation and shipping sectors used? Also, given the large uncertainties in the RF of many climate-relevant components, which in turn are critical for the total temperature impact (see also comment below) and hence the contribution of aviation and shipping, the authors need to provide information about the RF estimates (present day relative to pre-industrial) underlying their simulations. In particular, RF estimates specific to aviation and shipping – e.g., what is the indirect aerosol effect of shipping and aviation? And NOx-induced O3 and CH4 eff discuss the sectors contributions given that not only they, but also the rest of the world makes progress on emissions.

**Response:** The point about radiative forcings is an excellent one, and we have considerably expanded the text to discuss the radiative forcing estimates of both sectors as well as by species, added a new figure (Figure 3 - below), and compared our estimates to several previous studies. Given that we are modeling future forcing and temperature responses to aviation and shipping, we do not have present-day radiative forcing estimates, which would require historical $CO_2$ emissions for each sector in order to compute. However, we are still able to compare our future radiative forcing estimates to the literature based on knowledge of emissions inputs, and the fact that most species are short-lived.

The new discussion is as follows (P8:L28):

[revised manuscript text omitted]

Further, we note that there are not any particular parameterizations for the shipping and aviation sectors. In particular, we assume that all emissions take place at the surface of the Earth, which is a limitation of our analysis. We have highlighted this shortcoming of the model on P6:L29: *"We note that all emissions are treated as surface emissions. Aviation emissions in-flight occur at higher elevations, and this can affect atmospheric chemistry and radiation processes. For example, when sulfate is located above clouds, the radiative efficiency can be halved (less cooling); in contrast, the radiative efficiency of black carbon can be doubled (more warming) when it is located above clouds (Ocko et al. 2012). On the other hand, using more sophisticated climate models that can resolve horizontal and vertical granularities is often complicated by unforced internal variability that makes isolating the climate impact of relatively small radiative perturbations difficult if not impossible (Ocko et al. 2018)."*

[Figure]

**Comment 11:** Pg7, line 30: again, this is an example of where information about underlying RF is critical and should be compared with previous literature.

**Response:** We thank the referee for this emphasis on the need to discuss the underlying radiative forcings associated with the shipping and aviation sectors. As highlighted in the previous response, we have added a significant discussion of the radiative forcings derived by this study for the shipping and aviation sectors

and compared them to those from the literature. We appreciate how much this suggestion has strengthened our study.

**Comment 12:** Pg8, line 2: compare with other studies? E.g., Fuglestvedt et al. 2009.

**Response:** We thank the referee for the suggestion to substantiate our findings with a comparison to additional literature estimating the net climate impact of the shipping industry over the 21[st] century. We now compare the overall temperature trend associated with BAU emissions from the shipping sector presented by our own analysis and presented by Fuglestvedt et al. 2009, stating on P10:L15: *"This is also consistent with Fuglestvedt et al. (2009), which predicts that the accepted regulations in the shipping sector's emissions of sulfur dioxide and nitrogen oxides will lead to the sector having a net cooling effect for about 70 years, after which the sector switches to warming. Our analysis predicts a slightly more rapid shift to warming (after about 65 years in 2085), likely due to our inclusion of the warming climate pollutant black carbon which are not featured in the analysis by Fuglestvedt et al. (2009)."*

**Comment 13:** Pg8, line 19-21: for both sectors, the authors should also note that the calculations assume no change in geographical distribution of emissions. For non-CO2 emissions, location can be critical for the subsequent impact. E.g., Fuglestvedt et al. 2014; ES&T, Köhler et al. 2013 Atm. Environ; Frömming et al. 2012 JGR; Lund et al. 2017 ESD

**Response:** We appreciate the referee for bringing this detail to our attention. We now note that the geographical distribution of emissions is not included in our calculations for the relative climate impacts of each gas, utilizing the four studies from the literature suggested by the referee. We also highlight this point as a limitation of the MAGICC model overall, as it is not possible to look at vertical or horizontal changes in emissions density in a globally averaged model. We first highlight this challenge on P7:L13, stating "*Further, due to MAGICC's relative simplicity, parameters are averaged over large spatial scales. This is particularly important to acknowledge as recent literature has demonstrated that radiative forcings associated with the transport sector can differ based on the regional location at which the transport takes place (Berntsen et al. 2006; Fuglestvedt et al. 2014; Kohler et al. 2013; Fromming et al. 2012; Lund et al. 2017; Skowron et al. 2015), particularly for the impact of non-CO$_2$ emissions*" and again on P11:L3 "*We note that for both sectors, our calculations assume no change in the geographical distribution of emissions. Recent literature has demonstrated that the location of non-CO$_2$ emissions can have a large influence on their subsequent climate impact (Fuglestvedt et al. 2014; Kohler et al. 2013; Fromming et al. 2012; Lund et al. 2017; Skowron et al. 2015).*"

**Comment 14:** Pg8, line22 – onwards: As already pointed out in the major comment, I believe this result in misleading given the lack of treatment of contrail-cirrus. While it possible that the indirect aerosol effects of aviation sulfate and BC could be negative

enough to cause a net cooling, there is nothing in our current best understanding that suggests so. If included I think the authors should make a point of the missing effects at the very start of this paragraph not at the bottom, emphasizing that one should be careful not to read too much into this finding.

> **Response:** We appreciate the suggestion from the referee that we emphasize that contrail-cirrus is not included in these analyses before their results are presented. In both the new section on the radiative forcing estimates associated with the shipping and aviation sectors and the retained section on their related future warming, we have ensured that the exclusion of certain effects have been stated earlier in the text. We have added notes on P9:L17 stating, *"The net radiative forcing for international aviation emissions (note: not including impacts on contrails and cirrus clouds) is -1.4 mW m$^{-2}$ in 2020 and +62 mW m$^{-2}$ in 2100"* and on P11:L9 stating, *"However, the inclusion of non-CO$_2$ climate pollutant emissions does not yield a net cooling effect for several decades as they do with shipping, and reduces warming by end of century to 0.03 °C (note that we do not include here the impacts on contrails and cirrus clouds)."*

**Comment 15:** Pg8. line 31: again, the estimates of indirect aerosol RF should be included for comparison with e.g., Gettleman et al. 2013 GRL.

> **Response:** We thank the referee for the suggested reference and we have included a comparison of our indirect aerosol RFs with the results of this study as well as others for shipping (Righi et al. 2011). Additional text includes:
>
> P9:L1:*"Indirect aerosol effects from all species yield a radiative forcing of -32 mW m-2 in 2100."*
>
> P9:L15: *"Indirect effects of aerosols have enormous ranges in estimates in the literature (Righi et al. 2011 ), but we note that our estimate appears to be on the lower end."*
>
> P9:L22: *"Indirect aerosol effects from all species yield a radiative forcing of -10 mW m$^{-2}$ in 2100."*
>
> P9:L33: *"Gettelman and Chen (2013) conduct a more sophisticated assessment of the climate impact of aviation aerosols than what is presented here, and report an estimate of -46 mW m$^{-2}$ from combined sulfate direct and indirect effects; this is considerably larger than our estimate of -3 mW m$^{-2}$ in 2100."*

**Comment 16:** Pg9, lines 7-10: Skeie et al. 2009 included both indirect aerosol effects and contrailcirrus forcing – see their figure 2. Please correct or specify which indirect effects beyond there is included in this analysis.

**Response:** We appreciate the referee for highlighting this detail about Skeie et al. 2009. Upon further inspection of the article, the authors of Skeie et al. 2009 do include the indirect effects of nitrogen oxides via interactions with the lifetime of ozone and methane due to its impact on the lifetime of the hydroxyl radical, as well as the indirect aerosol effect of sulfur dioxide. However, to the best of our knowledge, they do not include the climate impact associated with the production of nitrate aerosols, which yields a significant cooling effect directly and indirectly. We have clarified this on P11:L30 as follows: "*first, our model includes indirect aerosol effects, particularly the climate impact associated with nitrogen oxides' production of nitrate aerosols, which yield negative forcings that are not considered in the analysis by Skeie et al. (2009).*"

**Comment 17:** Pg9, line 9: is the net NOx RF negative in Skeie et al. for shipping?

**Response**: The net $NO_x$ radiative forcing reported by Skeie et al. (2009) for shipping is positive. On page 6264 of Skeie et al. 2009, the authors write "the emission of $NO_x$ leads to a strong, short-lived positive $RF-O_3$, but also to a negative, long-lived forcing through changes in $CH_4$." While the long-lived, cooling effect of $NO_x$ is larger in the shipping sector than it is in the aviation sector, the net radiative forcing observed for the sector is still positive.

**Comment 18:** Pg11, line 4: the IPCC report on 1.5 degrees showed that there was a large difference between temperature response and time until reaching temperature thresholds between two simplified climate models. Uncertainties in the background temperature response affects the contribution from aviation and shipping, and should be discussed somewhere in the paper (perhaps the authors should consider a dedicated discussion section).

**Response:** This is a good point. We have added text to acknowledge this uncertainty: "*It is important to note that the background temperature response to other forcings (anthropogenic and natural) can affect the temperature responses to shipping and aviation. Therefore, even though they are ultimately subtracted out in our calculation, they do impact our results, and uncertainties in BAU emissions from other sectors and the resulting temperature effects need to be acknowledged*" (P8:L5)

We have also significantly expanded the discussion of model uncertainties in Section 2.3 in order to better address how uncertainties may affect our estimates of the contribution to future warming from shipping and aviation.

**Comment 19: Figure 4:** the authors should show also the CO2 only cases here, as in Figure 2, allowing the reader to assess the impact of the assumption that non-CO2 emissions are changed "proportionally".

**Response**: We appreciate the suggestion to clarify the impact of the assumption that non-$CO_2$ emissions are changed proportionally. We initially implemented this change to the graph, but found that the figure was more confusing than its original version. We have thus chosen to show only the all-products scenarios for what is now Figure 6.

**GENERAL COMMENTS**

**Comment 1:** A major criticism is that the results section is quite short and mentions very few previous studies. I would recommend to extend Section 3, adding more details and more citations, in particular concerning the role of the short-lived species. These compounds can be very relevant for the two sectors discussed here, as shown by several previous studies (see suggestions below). Also the uncertainties of the adopted simplified climate model in simulating the effects of short-lived species can be large and should be discussed.

> **Response:** We thank the reviewer for this observation, and have expanded Section 3, discussed more details of radiative forcing estimates and short-lived species, added information about uncertainties, added a sensitivity analysis regarding contrail/cirrus impacts, and added 27 studies to our references.
>
> Expanded discussion of uncertainties includes:
>
> on P6:L17, "*Whereas radiative impacts of well-mixed greenhouse gases (such as $CO_2$ and methane) are fairly well understood due to our knowledge of gas absorption, aerosol radiative effects are more complex and uncertain. This is due to spatial and temporal heterogeneity complicating observations; a variety of possible microphysical and optical properties based on varying sizes, shapes, structures, mixtures, and humidity levels; and interactions with clouds that can impact the lifetime and brightness of the clouds. Given that aerosols are quite relevant to both the aviation and shipping sectors (e.g. Unger et al., 2010), we include their direct and indirect effects in our simulations, noting that caution must be applied in interpreting the results.*"
>
> on P7:L13, "*Further, due to MAGICC's relative simplicity, parameters are averaged over large spatial scales. This is particularly important to acknowledge as recent literature has demonstrated that radiative forcings associated with the transport sector can differ based on the regional location at which the transport takes place (Berntsen et al. 2006; Fuglestvedt et al. 2014; Kohler et al. 2013; Fromming et al. 2012; Lund et al. 2017; Skowron et al. 2015), particularly for the impact of non-$CO_2$ emissions.*"
>
> on P6:L29, "*We note that all emissions are treated as surface emissions. Aviation emissions in-flight occur at higher elevations, and this can affect atmospheric chemistry and radiation processes. For example, when sulfate is located above clouds, the radiative efficiency can be halved (less cooling); in contrast, the radiative efficiency of black carbon can be doubled (more warming) when it is located above clouds (Ocko et al. 2012). On the other hand, using more sophisticated climate models that can resolve horizontal and vertical granularities is often complicated by unforced internal variability that makes*"

*isolating the climate impact of relatively small radiative perturbations difficult if not impossible (Ocko et al. 2018)."*

on P8:L5, *"It is important to note that the background temperature response to other forcings (anthropogenic and natural) can affect the temperature responses to shipping and aviation. Therefore, even though they are ultimately subtracted out in our calculation, they do impact our results, and uncertainties in BAU emissions from other sectors and the resulting temperature effects need to be acknowledged."*

**Comment 2:** In Sect. 2.1, scaling shipping SO2 emissions by a factor 7 to account for the IMO regulations in fuel sulfur content (FSC) only makes sense if the RCP8.5 dataset assumes a 3.5% FSC for the global shipping fleet. Is this really the case? The 3.5% cap was enforced in 2012, but it was 4.5% before and the RCPs scenarios start the projection in 2000. Moreover, according to the second IMO Study (Buhaug et al., 2009), the actual FSC in the global shipping fleet was on average 2.7% before the introduction of the IMO regulations. Therefore it could be that the FSC value assumed in RCP8.5 is lower than 3.5% and the scaling factor to get to 0.5% is lower than 7. Please check this.

**Response:** We thank the referee for pointing out these inconsistencies. We have reviewed the literature associated with the RCP database (Riahi et al. 2011) and recognize that the progressive reductions associated with the amendments to MARPOL Annex VI leading to an eventual 0.5% $SO_2$ emissions cap are indeed accounted for in the RCP8.5 database. All scenarios have been updated to return to the original sulfur emissions profiles provided by the database and are no longer altered based on the previous ratio.

**Comment 3:** End of Sect. 2.3: I understand that a full discussion of the model uncertainties is beyond the scope of this study, but I would at least briefly summarize which of them are the most significant for the results presented here.

**Response:** We thank the referee for this suggestion. We have significantly expanded the discussion of the model uncertainties in Section 2.3, particularly to discuss the potential impact of regional emissions, vertical differences in gas and aerosol concentrations throughout the atmosphere, our inability to accurately project future emissions over large spatial scales, and uncertainties in aerosol direct and indirect forcings. This new section runs from P6:L29 to P7:L23.

**Comment 4:** P8, L2: you may also want to compare with Lund et al. (Environ. Sci. Technol., 2012).

**Response:** We thank the referee for this suggestion. We have significantly expanded the comparison of our results to that found within the literature, including a comparison with Lund et al. 2012. The additional paragraph starts on P12:L9 and reads as follows: *"In the RCP scenarios presented by Lund et al.*

*(2012), shipping is projected to cause a cooling of between -0.02 and -0.04 °C by midcentury. Our analysis estimates that shipping is responsible for -0.03 °C in year 2050, which falls within this range. Further, the authors' findings are in agreement with those presented in this analysis through their observation of warming later in the century once the accumulating $CO_2$ emissions impact overruns the cooling impact of nitrous oxides and sulfur dioxide, particularly due to the reduced sulfur dioxide emissions associated with the implemented fuel regulations."* We have also included a comparison to Terrenoire et al. 2019 and Huszar et al. 2013 in order to assess whether our findings are consistent with multiple studies. Further, we have added information on the radiative forcing estimates from our model simulations, and compared the results to several previous studies.

**Comment 5:** P8, L16: there are a few studies simulating the aerosol indirect effect in low-sulfur shipping scenarios you may want to mention, for example Lauer et al. (Environ. Sci. Technol., 2009) and Righi et al. (Environ. Sci. Technol., 2011).

**Response:** We thank the referee for bringing these studies to our attention. We have highlighted this additional literature and its demonstration that low-sulfur shipping scenarios have the potential to reduce the indirect aerosol effect from shipping sulfur emissions. This helps strengthen our analysis regarding the net cooling reduction associated with implementation of the sulfur regulation. This section on P10:L29 now reads, *"Given that sulfur dioxide emissions—a precursor to the cooling pollutant sulfate—are projected to decrease significantly due to the sulfur fuel regulation newly adopted by IMO, sulfur dioxide from shipping contributes less significantly to cooling. Recent studies have demonstrated the potential for low-sulfur shipping scenarios to reduce the indirect aerosol effect from shipping sulfur emissions (Lauer et al., 2009; Righi et al. 2011)."*

**Comment 6:** P8, L26-27: the switch from cooling to warming is not evident in Fig. 2b. Does it occur before 2020? Please clarify.

**Response:** We thank the referee for bringing this concern to our attention. The switch from cooling to warming takes place around year 2024, but the line is very thick and this is hard to see. We have made the baseline outline a bit thinner in order to help with this, shown in what is now Figure 5a, reproduced below.

[Figure]

**Comment 7:** P8, L28-33: the issue of aviation effects of short-lived species should be discussed in more detail (see Lee et al., Atmos. Environ., 2010; or Grewe et al., Aerospace, 2018). There are several studies arguing for the effect of aviation soot on natural cirrus clouds (e.g., Penner et al., J. Geophys. Res., 2019) and some groups even argued for an effect on warm clouds (Gettelman and Chen, Geophys. Res. Lett., 2013; Righi et al., Atmos. Chem. Phys., 2013; Kapadia et al., Atmos. Chem. Phys., 2016). Can the simple climate model used here account for these effects?

> **Response:** This is a good point, and we have added discussion of these effects. The MAGICC model is currently not set up to account for effects beyond the standard first and second indirect effects of (all species of) aerosols on clouds. The text on P11:L18 now reads, *"We note that some studies have investigated the effect of aviation soot on natural cirrus clouds (Penner et al., 2019) or the effect on warm clouds (Gettleman and Chen, 2013; Righi et al., 2013; Kapadia et al., 2016). MAGICC does take into account indirect effects of soot, such as simplified parameterizations of impacts on cloud brightness and lifetime, but does not include more sophisticated treatments as analyzed in previous studies."*

**Comment 8:** P8, L29: what is the mechanism behind the cooling effect from nitrogen oxide? This gas can lead to the formation of ozone, which has a warming effect, but it also reduces methane lifetime, resulting in a cooling. Are these mechanisms included in the model?

**Response:** The net effects from $NO_x$ emissions are due to a combination of formation of ozone (positive forcing), reduction of methane (negative forcing), formation of nitrate aerosols (negative forcing), indirect effects of nitrate aerosols (negative forcing), and a cooler ocean suppressing $CO_2$ emission into the atmosphere (negative forcing). The overall effect is one of cooling as these mechanisms are all included in the model. We have clarified this in the text (P10:L27): "*The net cooling from nitrogen oxides arises from nitrate formation, indirect aerosol effects from nitrates, formation of tropospheric ozone, reduction of methane, and effects of the net forcings on the carbon cycle (cooling in the ocean suppresses $CO_2$ diffusion from the ocean into the atmosphere).*"

**Comment 9:** P8, L31-33: this is confusing, if you include the indirect aerosol effects, then you do address the impacts of aviation on cloudiness. You probably mean contrails and contrail-induced cloudiness here; please clarify and also add a citation to support the last statement in this sentence (warming effect).

**Response:** We see how this is confusing and we thank the referee for requesting this clarification. We did mean contrails and contrail-induced cloudiness. Considering that we have now included a sensitivity analysis of the climate impacts of contrails and contrail-induced cloudiness, the last part of this sentence has been removed.

**Comment 10:** End of Sect. 3.1: only one study is cited for comparison. It would be good to add more, possibly more recent, studies.

**Response:** This suggestion by the referee is greatly appreciated. We have significantly developed the comparison of our temperature results with other results from the literature, including a comparison to Lund et al., 2012; Terrenoire et al., 2019; and Huszar et al., 2013. Each of these studies is more recent than the originally cited Skeie et al. 2009. The comparison of our BAU warming results for international shipping and aviation with those from past literature now runs from P11:L23 to P12:L22. We have also added considerable text regarding radiative forcing estimates and compared our results to several previous studies as well.

**Comment 11:** Sect. 3.2: the role of short-lived pollutants in the aviation scenarios is not discussed at all. I understand that, unlike the shipping scenarios, the analyzed aviation scenarios do not distinguish between CO2 and non-CO2 species, but at least some qualitative considerations could be added here.

**Response:** We thank the referee for bringing this to our attention. We inserted a discussion at the end of Section 3.2 to underscore the fact that we do not consider the influence of reducing the non-$CO_2$ impact of the aviation sector like we do for the shipping sector. We have also outlined how the use of biofuels and changes to aircraft technology and air traffic management may impact estimated warming

mitigated through these interventions. This discussion begins on P14:L15 and states, "*Similarly, we do not consider the reduction of non-CO$_2$ climate pollutants emitted by the aviation sector in the mitigation scenarios. However, offsetting schemes such as CORSIA do implement the use of biofuels and aircraft technology and air traffic management improvements, both of which have the potential to impact future emissions of non-CO$_2$ climate pollutants and the density of contrail cirrus (Bock and Burkhardt 2019; Caiazzo et al., 2017; Burkhardt et al., 2018). While the influence of these changes on the non-CO$_2$ impact of international aviation is currently not well-estimated, their impact should be considered in future analyses as understanding develops..*"

**SPECIFIC COMMENTS**

**Comment 1:** P1, L14: please specify how much is this allowable warming.

> **Response:** We thank the referee for requesting this clarification. We specified the respective allowable warming for the two temperature thresholds. The clause now reads (on P1:L16): *"which is 12% and 24% of the "allowable warming" we have left to stay below the 2 °C or 1.5 °C thresholds (1.0 °C and 0.5°C) respectively."*

**Comment 2:** P1, L33: how does the time-frame affect the share of global CO2 emissions? Shouldn't it be rather given for a specific year?

> **Response:** We thank the referee for this clarifying question. We intended this sentence to reflect the sectors' share of total greenhouse gas emissions in CO$_2$-equivalents (including non-CO$_2$ emissions as well – which is why a time horizon was necessary). However, we have realized that this is confusing and overcomplicates our message. Therefore we have changed the sentence to reflect the sectors' share of CO2 emissions from energy sources. The sentence on P2:L1 now reads, "*While current emissions from international aviation and shipping account for around 4% of global energy-related CO$_2$ emissions (IMO, 2014; ICAO, 2019a; IEA, 2018), emissions from each sector are forecasted to increase anywhere from 200-400% (Lee 2018) and 50-250% (IMO, 2014) by midcentury, respectively, in the absence of effective policy.*"

**Comment 3:** P3, L18: I would change the title of Sect. 2.1, to make more clear that the baseline scenario is discussed here. I would also suggest to make two subsections of 2.1, to better separate aviation from shipping. The same would apply to 2.2. Another option would be to merge 2.1 and 2.2 in a single section on emissions, with two subsections for shipping and aviation, respectively.

> **Response:** We thank the referee for these suggestions in order to better organize our methodology description. We have changed the first section heading (section 2.1) to "Business-as-usual emissions from international bunkers" to reflect that it

is BAU emissions. We also included subsections within Sect. 2.1 to more clearly separate aviation from shipping.

**Comment 4:** P3, L27: "hold that level constant", I guess you are referring to the growth rate which is held constant, but that could be misunderstood as the actual emissions. I would be more explicit: "hold that growth rate constant".

**Response:** This scenario has changed in order to reflect the new data released by the International Civil Aviation Organization for future $CO_2$ emissions projections from the aviation sector. The description of this new scenario begins on P4:L5.

**Comment 5:** P4, L5: the RCP acronym should be explained.

**Response:** We appreciate this suggestion from the referee. We now explain on P3:L31 that RCP stands for Representative Concentration Pathways.

**Comment 6:** P4, L11: what do you mean by "all-forcings" BAU scenario?

**Response:** We thank the referee for this clarifying question. This first mention of the all-forcings scenario has been removed from the text, but we now provide clarifying language to define the all-forcings BAU scenario as "*the business-as-usual scenario including all natural and anthropogenic climate forcings*" on P4:L32.

**Comment 7:** P4, L29: there are more recent estimates, for example Burkhardt and Kärcher (Nature Clim. Change, 2011).

**Response:** We thank the referee for bringing more recent estimates to our attention. In our addition of a sensitivity analysis for the temperature impact of contrails and contrail induced cirrus cloudiness from P12:L23 to P13:L3, we reference five additional studies (namely Lee et al., 2009; Schumann et al., 2015; Brasseur et al., 2016; Bock and Burkhardt, 2016, and Burkhardt et al., 2019), each of which are more recent than Sausen et al., 2005.

**Comment 8:** P6, L24: please provide a reference for these relationships.

**Response:** We thank the referee for this suggestion. This relationship is a ratio provided by the EDGAR database as outlined in Crippa et al. 2016. We have provided the reference more explicitly as instructed on P4:L31.

**Comment 9:** P6, L26: please replace "gas" by "species" or "compound", since also aerosols are considered here.

**Response:** We appreciate the referee for pointing out this oversight. We replace the word "gas" with "species" as suggested on P6:L25.

**Comment 10:** P7, L4: 2100 - 1765 + 1 = 336 years (?)

**Response:** We thank the referee for bringing this ambiguity to our attention. The number 335 represents the number of integrations we are performing, rather than the number of years. This was unclear in our original language, and we have changed the language to better reflect our intended meaning. The sentence on P7:L25 now reads *"We run 335 year-to-year integrations from year 1765 to 2100 for a set of 14 different simulations."*

**Comment 11:** P11, L13: it might be worthwhile to cite Fuglestvedt et al. (Environ. Sci. Technol., 2009) in this context.

**Response:** We thank the referee for this suggestion. We added in this citation to further support the statement on P15:L8.

**Comment 12:** P11, L14-16: since this is the main motivation behind this work, I would suggest putting this sentence also in the introduction.

**Response:** We thank the referee for this suggestion. We have inserted a similar sentence into the introduction in order to provide this compelling context in that section as well on P1:L13 which reads, *"Given that the global average temperature has already risen 1 °C above preindustrial levels, there exists only 1.0 °C or 0.5°C of additional "allowable warming" left to stabilize below the 2 °C or 1.5 °C thresholds, respectively. We find that if no actions are taken, $CO_2$ emissions from international shipping and aviation may contribute roughly equally to an additional combined 0.12 °C to global temperature rise by end of century—which is 12% and 24% of the "allowable warming" we have left to stay below the 2 °C or 1.5 °C thresholds (1.0 °C and 0.5°C) respectively."*

**Comment 13:** Figure 1: the acronym MMT should be explained. Also, this is a non-SI unit: I would use Tg or Gg instead.

**Response:** We appreciate this suggestion from the referee. We have changed these units to the equivalent SI unit, Tg.

**Comment 14:** Figure 3a is discussed before Figure 2b. You could think about grouping the plots by sector since this reflects the way they are presented in the text

**Response:** We thank the referee for this suggestion. We now group the plots by sector, as suggested. The new Figure 4 and Figure 5 are reproduced below, for reference.

[Figure]

**Comment 15:** Figure 4: it is hard to distinguish the lines for the different scenarios, since very similar colors are used for them.

**Response:** We appreciate this suggestion from the referee. We have changed the colors of the figure in order to better distinguish between the different scenarios. The updated figure (now Figure 6) is reproduced below, for reference. Note that the colors for Figure 7 (also reproduced below) have also been changed in order to keep the color scheme consistent for each policy scenario.

[Figure]

[Figure]

**(a) SHIPPING**

**(b) AVIATION**

Avoided Warming (°C) in 2100

[revised manuscript text omitted]

As with climate models of any complexity level, there are limitations in our knowledge of climate and carbon cycle processes, radiative forcings, and especially indirect aerosol effects, which introduce uncertainties within the model. While MAGICC uses several calibration methods to determine its parameters from a large collection of sophisticated models, the comprehensive models will pass along their own uncertainties to MAGICC. Further, due to MAGICC's relative simplicity, parameters are averaged over large spatial scales. This is particularly important to acknowledge as recent literature has demonstrated that radiative forcings associated with the transport sector can differ based on the regional location at which the transport takes place (Berntsen et al. 2006; Fuglestvedt et al. 2014; Kohler et al. 2013; Fromming et al. 2012; Lund et al. 2017; Skowron et al. 2015), particularly for the impact of non-$CO_2$ emissions.

MAGICC also does not account for vertical differences in gas and aerosol concentrations throughout the atmosphere, instead treating all emissions as surface emissions. This may not be a major issue for the surface-level emissions occurring in the shipping industry, but becomes more complex when considering emissions from aviation at various altitudes. For example, when sulfates are emitted above clouds, the radiative forcing is reduced; in contrast, the emission of black carbon above the cloud level would be enhanced (Ocko 2012). While more sophisticated climate models may be able to include this horizontal and vertical granularity, their internal variation make isolating the climate impact of relatively small sectoral perturbations virtually impossible (Ocko 2018). The ability of MAGICC to identify these climate responses make it a compelling choice for this type of small scale analysis.

Other major sources of uncertainty stem from the innate inability to perfectlyaccurately project future emissions due to uncertainties in both the human and the climate components of prediction. All mitigation scenarios are compared to an estimated baseline, and the social and economic data utilized in order to inform this estimated baseline cannot be expected to perfectly match the unpredictable nature of human action. Further, the large spatial scales and parameterizations involved in climate modeling contribute to some degree of uncertainty. A full discussion of model uncertainties can be found in Meinshausen et al. (2011a).

**2.4 Climate model simulations**

We run 335 -year-to-year integrations from year 1765 to 2100 for a set of 17 14 different simulations. These simulations are comprised of five BAU pathways and 12 nine mitigation pathways based on current and potential policy scenarios within the international aviation and shipping sectors. For future emissions from sectors other than international aviation and shipping, we use RCP8.5 emissions data, but the climate impacts are subtracted out as described below.

The five BAU scenarios account for the warming impacts due to: all natural and anthropogenic forcings; isolation of the $CO_2$ emissions from international shipping; isolation of the $CO_2$ emissions from international aviation; isolation of the $CO_2$, black carbon, methane, nitrogen oxides, sulfur dioxides, organic carbon, and carbon monoxide emissions from international shipping; and isolation of the $CO_2$, black carbon, nitrogen oxides, sulfur dioxide, organic carbon, and carbon monoxide emissions from international aviation. The  nine mitigation simulations account for the future emissions pathways for the  nine policy scenarios outlined in Table 1.

~~MAGICC also does not account for vertical differences in gas and aerosol concentrations throughout the atmosphere, instead treating all emissions as surface emissions. This may not be a major issue for the surface-level emissions occurring in the shipping industry, but becomes more complex when considering emissions from aviation at various altitudes. For example, when sulfates are emitted above clouds, the radiative forcing is reduced; in contrast, the emission of black carbon above the cloud level would be enhanced (Ocko 2012). While more sophisticated climate models may be able to include this horizontal and vertical granularity, their internal variation make isolating the climate impact of relatively small sectoral perturbations virtually impossible (Ocko 2018). The ability of MAGICC to identify these climate responses make it a compelling choice for this type of small scale analysis.~~

In order to isolate sector emissions in each BAU and mitigation scenario, we subtract the total emissions of all gases and aerosols associated with each sector from the total RCP8.5 emissions of all gases and aerosols in the all-forcing scenario driven by all natural and anthropogenic forcings (Eq. 1). The annual average mean surface temperature changes from these emissions profiles are subtracted from the temperature changes in the all-forcings scenario in order to determine the contribution to future temperature change from each sector (Eq. 2). It is important to note that the background temperature response to other forcings (anthropogenic and natural) can affect the temperature responses to shipping and aviation. Therefore, even though they are ultimately subtracted out in our calculation, they do impact our results, and uncertainties in BAU emissions from other sectors and the resulting temperature effects need to be acknowledged.

$$Emissions_{all-forcings\ without\ sector} = Emissions_{all-forcings} - Emissions_{sector} \qquad\qquad (1)$$

$$\Delta T_{sector} = \Delta T_{all-forcings} - \Delta T_{all-forcings\ without\ sector\ emissions} \qquad\qquad (2)$$

The comparison of each sector's baseline scenario to its respective mitigation scenarios are analyzed independently from other potential mitigation efforts that may occur in the future. Thus, isolating the temperature impacts of a given mitigation scenario does not mandate that all other anthropogenic emissions continue unabated. The same methodology can be used to isolate temperature changes due to individual gases or aerosols for each sector.

$$Emissions_{all-forcings\ without\ sector} = Emissions_{all-forcings} - Emissions_{sector} \qquad\qquad (1)$$

$$\Delta T_{sector} = \Delta T_{all-forcings} - \Delta T_{all-forcings\ without\ sector\ emissions} \qquad\qquad (2)$$

MAGICC also does not account for vertical differences in gas and aerosol concentrations throughout the atmosphere, instead treating This may not be a major issue for the surface level emissions occurring in the shipping industry, but becomes more complex when considering emissions from aviation at various altitudes.

**3 Results**

[revised manuscript text omitted]

**(a)**

**(b)**

[Figure]

AVIATION BAU, CO₂ ONLY

AVIATION BAU, ALL POLLUTANTS

[Figure]

Contribution of Future Emissions to Surface Air Temperature Change (°C)

**Figure** 5: Contribution of future emissions to surface air temperature change in °C associated with business-as-usual emissions starting in 2020 and continuing through the end of the century from international aviation. Future temperature impacts is presented a)  for emissions of $CO_2$ only (thin line) and all pollutants (thick line), and b) by the contribution of individual pollutants in 2100.

[Figure]

[Figure]

**Figure 4 6: Surface air temperature changes associated with various policy scenarios for emissions mitigation in international a) shipping and b) aviation. Each business-as-usual scenario presents the contribution to future surface air temperature from the emissions of all climate pollutants starting in 2020 and continuing through the end of the century.**

[Figure]

(a) SHIPPING

(b) AVIATION

Avoided Warming (°C) in 2100

[Figure]

**Figure 5 7: Avoided warming in year 2100 associated with various policy scenarios for emissions mitigation in international a) shipping and b) aviation.**

---

## Author Response (AR2)

**RESPONSE TO THE REFEREE COMMENTS**
**DURING THE CORRECTION STAGE**

**Manuscript Ref: acp-2019-126**

**Climate benefits of proposed carbon dioxide mitigation strategies for international**

**shipping and aviation**

Catherine C. Ivanovich, Ilissa B. Ocko, Pedro Piris-Cabezas, and Annie Petsonk

We sincerely appreciate the careful reviews and helpful suggestions provided by the Reviewers, and thank the Reviewers and the Editor for their time. We have made changes to the manuscript in response to the three technical corrections posed by Referee #2.

Below are the point-by-point responses to the comments and the subsequent changes to the paper (reviewer comments in blue, responses in black).

**Responses to Anonymous Referee #2:**

**Comment 1:** P2 L25: please explain "CO2e".

**Response:** The needed additional clarity to explain what is meant by $CO_2e$ is a great point made by the referee. We have developed the description as follows, "*Many studies aggregate the climate impacts of each sector through the use of $CO_2$-equivalence ($CO_2e$) (Lee et al., 2010; Lee, 2018; Eyring et al., 2010; Azar and Johansson, 2012). While the use of this simple metric attempts to describe the global warming intensity associated with the emission of multiple greenhouse gases, it does not account for continuous emissions nor convey warming impacts over time (Ocko et al., 2017)*" on P2:L25.

**Comment 2:** P4 L1: "emissions cap" --> "emissions cap in 2020".

**Response:** We appreciate this suggested edit. The sentence now reads: "*These projections do include progressive reductions to sulfur dioxide emissions associated with the amendments to MARPOL Annex VI, which leads to a 0.5% sulfur dioxide emissions cap in 2020*" on P3:L35.

**Comment 3:** P9 L13: "0.2?" looks not correct.

**Response**: We thank the referee for bringing this error to our attention. We have deleted the question mark and corrected the referenced value, and the sentence now reads, "*
[revised manuscript text omitted]